# From Global Assessment to Local Selection: Efficiently Solving Traveling Salesman Problems of All Sizes

## Abstract

The Traveling Salesman Problem (TSP) is a well-known combinatorial optimization problem with broad real-world applications. Recent advancements in neural network-based TSP solvers have shown promising results. Nonetheless, these models often struggle to efficiently solve both small- and large-scale TSPs using the same set of pre-trained model parameters, limiting their practical utility. To address this issue, we introduce a novel neural TSP solver named GELD, built upon our proposed broad global assessment and refined local selection framework. Specifically, GELD integrates a lightweight Global-view Encoder (GE) with a heavyweight Local-view Decoder (LD) to enrich embedding representation while accelerating the decision-making process. Moreover, GE incorporates a novel low-complexity attention mechanism, allowing GELD to achieve low inference latency and scalability to larger-scale TSPs. Additionally, we propose a two-stage training strategy that utilizes training instances of different sizes to bolster GELD's generalization ability. Extensive experiments conducted on both synthetic and real-world datasets demonstrate that GELD outperforms seven state-of-the-art models considering both solution quality and inference speed. Furthermore, GELD can be employed as a post-processing method to exchange affordable computing time for significantly improved solution quality, capable of solving TSPs with up to 744,710 nodes without relying on divide-and-conquer strategies.

## 1 Introduction

The Traveling Salesman Problem (TSP) is one of the most well-known Combinatorial Optimization Problems (COPs) and has extensive real-world applications (Ha et al., 2018). Due to the practical significance of TSP, many exact, approximate, and heuristic algorithms have been developed over the years. Recently, advances in deep learning have led researchers to develop Neural Networks (NNs) as a kind of viable solvers for TSPs (Wu et al., 2024). Although theoretical guarantees for such networks remain elusive, they tend to produce near-optimal solutions in practice, offering faster inference speed and better generalization than conventional TSP solvers (Bengio et al., 2021).

Neural TSP solvers often demonstrate excellent performance when trained and tested on small-scale TSPs (e.g., around 100 nodes) (Kwon et al., 2020). However, existing models generally face the following four key limitations: 1) Generalizing pre-trained models to TSPs of different sizes often results in substantial performance degradation. This limitation poses a major obstacle towards deploying these models because real-world TSPs often involve tasks of varying sizes; 2) The quadratic time-space complexity ($\mathcal{O}(n^2)$, where $n$ denotes the number of nodes in the underlying TSP) of the standard attention mechanism commonly used in neural TSP solvers restricts their applicability to large-scale TSPs (e.g., over 1,000 nodes); 3) Further elevation of solution quality, e.g., via sacrificing computing time, is challenging because the NN used in neural TSP solvers typically serves as fixed mapping functions from node features to TSP solutions (Xiao et al., 2024b); and 4) While models based on the Divide-and-Conquer (D&C) strategy perform well when solving large-scale TSPs (Zheng et al., 2024), they may fail to provide valuable insights for solving other COPs, such as the Job Shop Scheduling Problem (JSSP), which requires rigid sequential execution and is not easily divisible. Therefore, in this work, we investigate the following research question:

*Can a unified pre-trained model, not based on D&C, effectively solve both small- and large-scale TSPs in a short time period while further elevating solution quality at the cost of affordable time?*

To answer this research question, we introduce GELD, a novel model that integrates a Global-view Encoder (GE) and a Local-view Decoder (LD) to efficiently solve TSPs. Firstly, GELD is built upon our proposed broad global assessment and refined local selection framework (see Section 3.3). Specifically, GELD employs a lightweight GE to capture the topological information across all nodes in the underlying TSP, paired with a heavyweight LD to autoregressively select the most promising node within a local selection range. This dual-perspective approach enriches the embedding representation by integrating both global and local insights while accelerating the selection process by confining the decision space to a smaller, more relevant subset, thereby improving both efficiency and generalization. Secondly, to reduce model complexity and further accelerate inference, we propose a novel Region-Average Linear Attention (RALA) mechanism within GE which operates with $\mathcal{O}(n)$ time-space complexity. RALA partitions the nodes in the underlying TSP into regions and facilitates efficient global information exchange through regional proxies, allowing GELD to solve TSPs in a short time period and scale effectively to larger instances. Thirdly, to further elevate solution quality, we incorporate our proposed idea of diversifying model inputs (see Section 3.4) into GELD's architectural design, enabling the model to function not only as a TSP solver but also as a powerful post-processing method to efficiently exchange affordable computing time for improved solution quality. Finally, to ensure GELD's robustness across TSPs of all sizes, we propose a two-stage training strategy, incorporating instances of varying sizes. This approach further strengthens the model's generalization capability, allowing it to solve TSPs efficiently with the same set of pre-trained model parameters.

To evaluate the effectiveness of GELD, we conduct extensive experiments on both synthetic and widely adopted benchmarking real-world datasets. The results demonstrate that GELD outperforms seven State-of-the-Art (SOTA) models considering both solution quality and inference speed. Furthermore, as a post-processing method, GELD not only significantly enhances the solution quality of baseline models with insignificant additional computing time, but also effectively solves extremely large TSPs (up to 744,710 nodes) when integrated with conventional heuristic algorithms. Our findings strongly suggest that GELD is by far the most SOTA model for solving TSPs.

The key contributions of this work are as follows.

i) To the best of our knowledge, GELD is the first unified model with a single set of pre-trained parameters that effectively solves TSPs of all sizes while efficiently enhancing solution quality.

ii) We propose a novel low-complexity encoder-decoder backbone architecture for GELD, enabling low-latency problem-solving and scalability to larger TSP instances.

iii) We propose a two-stage training strategy that utilizes instances of varying sizes to enhance GELD's generalization ability.

iv) We show the effectiveness of GELD both as a standalone TSP solver and as a powerful post-processing method that exchanges time for solution quality by conducting extensive experiments.

## 2 RELATED WORK

In this section, we review the NN-based methods for solving TSPs and then introduce several recent endeavors aimed at enhancing model generalization.

### 2.1 NEURAL NETWORK-BASED TSP SOLVERS

NN-based methods have shown promising results in solving TSPs and can be broadly classified into the following two categories: 1) Neural construction methods. These methods produce TSP solutions either autoregressively (majority) (Kool et al., 2019; Jin et al., 2023) or in a one-shot manner (minority) (Xiao et al., 2023; Min et al., 2023). For instance, Kool et al. (2019) proposed a well-known Attention Model (AM) for solving TSPs. Moreover, numerous studies extended AM and achieved better solution quality (Kim et al., 2022; Kwon et al., 2021; Chalumeau et al., 2023), with POMO (Kwon et al., 2020) being the most representative model. Recently, Xiao et al. (2024a) introduced the GNARKD method, which distills autoregressive models into those capable of producing solutions in a one-shot manner, significantly reducing inference time. 2) Neural improvement

methods. These methods start with initial solutions and employ deep learning techniques, such as pre-trained NNs to guide or assist the optimization of heuristics to iteratively improve the solutions (Li et al., 2023). In this line of research, local search (Hudson et al., 2022; Ma et al., 2023) and evolutionary computation (Ye et al., 2023; Kim et al., 2024) algorithms are often utilized.

Despite progress in both categories, these methods typically operate independently. To the best of our knowledge, there does not exist a unified approach capable of both producing and improving TSP solutions. To fill in this gap, in this paper, we propose a unified model that serves as both a standalone TSP solver and a post-processing method to further elevate solution quality.

## 2.2 GENERALIZATION OF NEURAL TSP SOLVERS

Early studies on neural TSP solver primarily focused on small-scale instances, which limited their applicability to practical and larger-scale scenarios. Recent efforts have sought to extend pre-trained models to larger-scale TSPs, often employing D&C strategies (Fu et al., 2021; Li et al., 2021; Cheng et al., 2023; Hou et al., 2023; Pan et al., 2023; Ye et al., 2024; Yu et al., 2024). These models decompose a large-scale problem into multiple smaller sub-problems, solve them individually or in parallel, and then combine the solutions of these sub-problems to construct the complete solution for the original problem. While effective for large-scale TSPs, D&C-based methods may be less suitable for more complex COPs such as JSSP, because decomposing such problems is often intractable using a unified model or strategy (Luo et al., 2024). Additionally, D&C may overlook correlations between sub-problems, potentially leading to suboptimal solutions (Luo et al., 2024).

Beyond D&C-based neural TSP solvers, alternative learning paradigms, such as diffusion models (Sun & Yang, 2023), have shown excellent performance in solving large-scale TSPs. Among these non-D&C-based neural TSP solvers, BQ (Drakulic et al., 2023) and LEHD (Luo et al., 2023) demonstrated promising results. By leveraging the recursion nature of COPs, BQ yielded notable results not only in solving large-scale TSPs but also in solving other challenging divisible COPs, such as JSSP (Pirnay & Grimm, 2024). However, these models struggle to solve TSPs exceeding 1,000 nodes and require significant computing time (see Table 1), limiting their real-world applicability.

To improve the practicality of neural TSP solvers and provide insights for solving other COPs, we aim to effectively solve both small- and large-scale TSPs without relying on D&C strategies.

# 3 PRELIMINARIES

This section first details the TSP setting and the autoregressive mechanisms used in neural TSP solvers. Next, we identify potential generalization issues in neural TSP solvers and outline the motivation behind the framework design of GELD. Finally, we review existing operations that exchange computing time for elevated solution quality and discuss the rationale for diversifying model inputs.

## 3.1 TSP SETTING

Our research focuses on the most fundamental Euclidean TSP due to its importance and prevalence in various application domains (Applegate et al., 2007; Qiu et al., 2022). We denote a TSP-$n$ instance as a graph with $n$ nodes in the node set $V$, where node $x_i \in \mathbb{R}^{n \times d}$ denotes the $d$-dimensional node coordinates. We define a TSP tour as a permutation of $n$ nodes denoted by $\pi = \{\pi_1, \pi_2, ..., \pi_n\}$, where $\pi_i \neq \pi_j, \forall i \neq j$. The length of a TSP tour $\pi$ is defined as follows:

$$\mathrm{L}(\pi) = \mathrm{d}(x_{\pi_1}, x_{\pi_n}) + \sum_{i=1}^{n-1} \mathrm{d}(x_{\pi_i}, x_{\pi_{i+1}}), \tag{1}$$

where $\mathrm{d}(x_{\pi_i}, x_{\pi_j})$ denotes the Euclidean distance, measured without considering direction, between nodes $\pi_i$ and $\pi_j$. The goal is to find a feasible solution $\pi^*$ that minimizes the length $\mathrm{L}(\pi^*)$.

## 3.2 AUTOREGRESSIVE NEURAL TSP SOLVERS

Autoregressive models are commonly employed to solve TSPs following the Markov Decision Process (MDP). At each step $t$ of the MDP, the model whose parameters are denoted as $\theta$, takes an action $a_t$ based on the previously taken actions $a_{1:t-1}$ to choose an unvisited node, until the tour is completed. Given a TSP instance $s$, this process can be factorized into a chain of conditional probabilities as follows:

$$p_\theta(\pi|s) = \prod_{t=1}^{n} p_\theta(a_t|a_{1:t-1}, s). \tag{2}$$

### 3.3 Generalization Issues

Effectively generalizing across TSPs of varying sizes is a crucial capability for NN-based models (Joshi et al., 2022; Zong et al., 2022). This task is challenging due to the explosive growth in the feasible solution space ($\mathcal{O}(n!)$) as the size $n$ increases. In autoregressive neural TSP solvers, larger-size instances lead to both increased MDP steps and an expanded decision space (i.e., available nodes) at each step (see (2)). To better deal with these issues, we propose to confine the decision space at each step to a limited range. Our strategy has certain resemblance to the recent INViT model (Fang et al., 2024), which utilizes multiple local views to solve large-scale TSPs. While INViT excels in solving large-scale TSPs, its exclusive focus on local information results in suboptimal performance on smaller-scale ones (see Table 1). Conversely, models such as ELG (Gao et al., 2024), which integrate both global and local views, tend to prioritize local information for decision-making without reducing the decision space. Consequently, these models still face challenges in effectively solving large-scale TSPs (see Table 2).

Unlike previous approaches, we introduce a novel *broad global assessment and refined local selection* framework in this paper, which draws inspiration from common decision-making processes in daily life: We often survey adequate relevant information broadly before carefully selecting the most promising option from several candidates. When applied to solve COPs, this framework involves an initial rough assessment of the entire problem, followed by a zoomed-in focus on the promising candidates, and selection of the most promising one as the action at each decision step. Building upon this idea, we aim to generalize our model to effectively solve TSPs of all sizes.

### 3.4 Methods of Exchanging Time for Further Elevated Solution Quality

Neural TSP solvers often utilize a greedy strategy, selecting the node with the highest probability at each MDP step. While computationally efficient, this approach often results in suboptimal solutions (Hottung et al., 2022). To improve solution quality, researchers have proposed various methods, often at the expense of increased computing time. These methods can be broadly categorized into the following two types: 1) Producing multiple candidate solutions utilizing techniques such as data augmentation (Geisler et al., 2022), multiple rollouts (Kwon et al., 2020; Hottung et al., 2024), and various search methods (Choo et al., 2022; I. Garmendia et al., 2024); and 2) Employing post-processing techniques, such as 2-opt (Sun & Yang, 2023), monte carlo tree search (Xia et al., 2024), and Re-Construction (RC) (Luo et al., 2023; 2024; Ye et al., 2024) to improve the quality of initial solutions. Given the versatility and efficiency of these approaches, we primarily employ Beam Search (BS) and RC to balance computing time and solution quality.

BS is a breadth-first search method with a predefined width $B$ (Kool et al., 2019). It begins with the starting node and incrementally expands the tour by evaluating $B$ potential successors. At each step, BS retains the top-$B$ sub-tours based on their cumulative logarithmic probabilities.

After obtaining initial solutions, RC randomly selects sub-solutions, reintegrates their node features into the model, and generates new sub-solutions using a greedy strategy. If these new sub-solutions are of higher quality, they replace the current ones. Importantly, RC is fundamentally distinct from the D&C strategy which decomposes a large problem into multiple smaller sub-problems—a process that can be particularly challenging for certain COPs such as JSSP. Instead, RC exploits the property that the optimal solution of COPs comprises optimal sub-solutions. By enhancing the quality of these sub-solutions, the overall solution quality is improved, making such an approach applicable to a broader range of scenarios. Furthermore, when multiple sub-solutions are processed in parallel, referred to as Parallel RC (PRC) (Luo et al., 2024), this parallel approach yields promising results in effectively exchanging computing time for further elevated solution quality.

We attribute the effectiveness of RC to the diversification of model inputs. The rationale behind this is as follows: RC improves solution quality by generating different sub-solutions, which essentially expands the search space. However, NNs are often treated as fixed mapping functions from inputs to outputs. If the model's inputs remain relatively unchanged, the search space is restricted, leading to relatively fixed outputs and limited solution quality improvement possibilities. Therefore, we deem that increasing the diversification of model inputs may enhance the effectiveness of RC. Based on this rationale, we impose the need for diversified inputs during the RC process in our model architectural design (see Section 4.1). We present the detailed RC process in Appendix A.1.

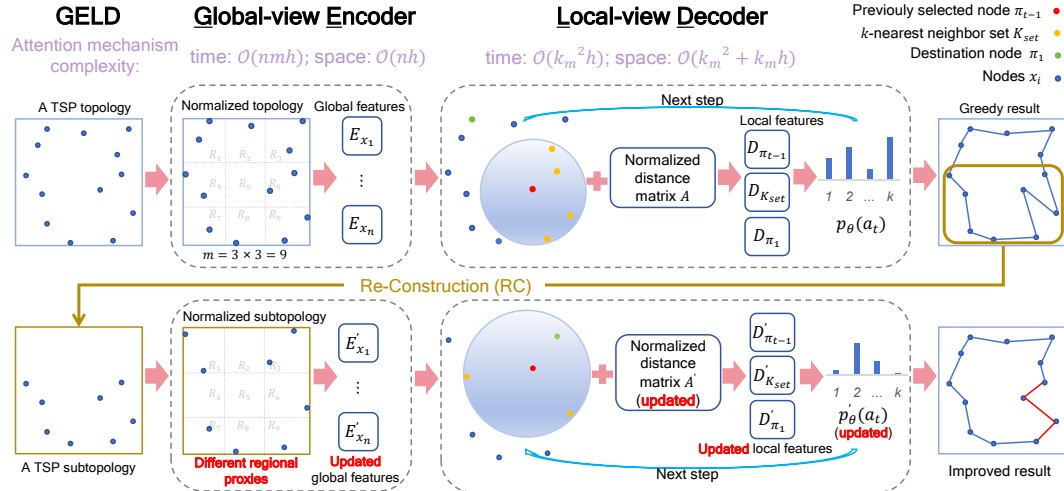

Figure 1: Framework of our proposed GELD, which incorporates a low complexity architecture with a global-view encoder and a local-view decoder. Furthermore, the effectiveness of RC is improved by considering the need for diversified inputs in the model architectural design.

## 4 GELD: GLOBAL-VIEW ENCODER AND LOCAL-VIEW DECODER

This section introduces a novel neural TSP solver named GELD. We detail the model architecture and training strategy of GELD in the following subsections.

### 4.1 ARCHITECTURE OF GELD

In alignment with the broad global assessment and refined local selection framework, we adopt an encoder-decoder architecture. The encoder captures the topological information across all nodes in the underlying TSP with a global view (*global assessment*), while the decoder employs a local perspective to autoregressively generate the probability distribution for selecting the next node at each step of the MDP (*local selection*). We present the overall framework of GELD in Figure 1.

**Global-view Encoder.** To capture global information in the TSP, we account for several distribution patterns, such as the clustered distribution (Bossek et al., 2019), which may only occupy a subset of the graph. Before identifying the patterns, we first normalize the node coordinates $x$ as follows:

$$\phi(x) = \frac{x - \min_{x_i \in V}(x_i)}{\max_{x_i, x_j \in V}(x_i - x_j)}. \tag{3}$$

Furthermore, during the RC process, the normalization operation alters the node coordinates according to the node changes in node set $V$, which consists of (different) nodes derived from randomly selected sub-solutions, thereby modifying the model input and enhancing the efficacy of RC.

Then, we linearly project the normalized coordinates into an $h$-dimensional embedding as follows:

$$E = \phi(x)W + b, E \in \mathbb{R}^{n \times h}, \tag{4}$$

where $W$ and $b$ denote the learnable parameters of weights and biases, respectively.

In alignment with the *broad global assessment* aspect of the proposed framework, which involves broadly surveying the relevant TSP information, we utilize *a single* (*broad*) attention layer to extract *global* features of nodes. Notably, extracting these global features presents challenges because it requires meeting the following three criteria: 1) Comprehensive coverage of all node information to enable interaction among nodes and facilitate global information transfer; 2) Low computational complexity to ensure scalability to larger-scale TSPs; and 3) The ability to obtain global information in a vague manner, allowing for effective diversifying model inputs during the RC process.

Existing models often adopt the standard attention mechanism (Vaswani et al., 2017) to facilitate global information transfer, which aids in mapping a query $Q = EW$ to an output using a set of key-value pairs $K = EW$ and $V = EW, Q, K, V \in \mathbb{R}^{n \times h}$ as follows:

$$E = \text{Softmax}(QK^T)V, E \in \mathbb{R}^{n \times h}. \tag{5}$$

While the standard attention mechanism delivers strong performance, its quadratic complexity, specifically the time complexity of $\mathcal{O}(n^2 h)$ and the space complexity of $\mathcal{O}(n^2 + nh)$, limits a model's scalability to larger-scale instances.

To meet the aforementioned three criteria, we propose Region-Average Linear Attention (RALA) that captures global node features with a reduced computational complexity. We present the detailed computation process of RALA in Figure 2. Specifically, we first partition all nodes into $m$ regions according to the normalized node coordinates, denoted as $R_1, \ldots, R_m$. Here, $m = m_r \cdot m_c$ and $m \ll n, h$, where $m_r, m_c \in \mathbb{Z}^+$ denote the predefined numbers of rows and columns for partitioning, respectively. The derived hyperparameter $m$ controls the granularity of the regional view: a larger value of $m$ may capture more insights of local regions but increases complexity. Then, we employ regional proxies to facilitate global information exchange among all nodes, thereby meeting the first aforementioned criterion. We compute the embedding of each regional proxy $P_i$ by averaging the query embedding $Q$ of all nodes in this region as follows:

$$P_i = \begin{cases} \frac{1}{n_{R_i}} \sum Q_{x_j}, x_j \in R_i, & \text{if } n_{R_i} > 0, \\ 0_{1 \times h}, & \text{otherwise,} \end{cases} i \in \{1, \ldots m\}, P \in \mathbb{R}^{m \times h}, \quad (6)$$

where $n_{R_i}$ denotes the number of nodes in region $R_i$ and $Q_{x_i} \in \mathbb{R}^{1 \times h}$ denotes the embedding of node $x_i$ in the query $Q$.

Next, we compute the node's query weight score for each region as follows:

$$Q_w = \text{Softmax}(QP^T), Q_w \in \mathbb{R}^{n \times m}. \quad (7)$$

Similarly, we compute the regional proxy's key weight score for each node as follows:

$$K_w = \text{Softmax}(PK^T), K_w \in \mathbb{R}^{m \times n}. \quad (8)$$

Finally, we update the node features to facilitate the global information transfer as follows:

$$E = Q_w(K_w V), E \in \mathbb{R}^{n \times h}. \quad (9)$$

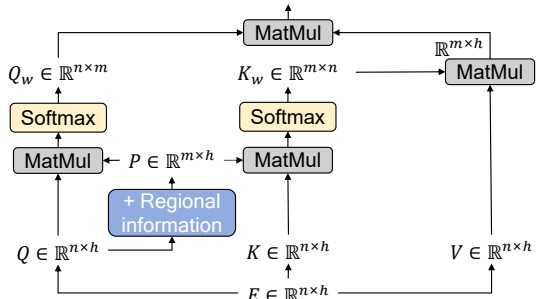

Figure 2: Computation process of RALA.

Unlike the quadratic complexity of the standard attention mechanism, our proposed RALA achieves a time and space complexity of $\mathcal{O}(nmh)$ and $\mathcal{O}(nh)$, respectively, without introducing any additional learnable parameters. This efficiency makes RALA meet the second aforementioned criterion, capable of solving large-scale instances efficiently. Furthermore, during the RC process, the introduction of normalization operations (see (3)) leads to nodes being assigned to different regions for RALA execution, as illustrated in Figure 1. The diversification in regional proxies updates the global features and then enhances the effectiveness of RC, meeting the third aforementioned criterion.

**Local-view Decoder.** In alignment with the *refined local selection* aspect of the proposed framework, which selects the most promising option from several candidates, we utilize *multiple (refined)* attention layers within the local-view decoder. Following the decoder design adopted in LEHD (Luo et al., 2023) and BQ (Drakulic et al., 2023), we select the most promising node $\pi_t$ from a candidate set based on the information from the previously selected node $\pi_{t-1}$ and the destination node $\pi_1$ at MDP step $t$. Unlike LEHD and BQ that consider all available nodes as candidates, we restrict the candidate set to the available $k$-nearest neighbors $K_{set}$ of node $\pi_{t-1}$ (i.e., *local selection*), where $k = \min\{k_m, n_t\}$, with hyperparameter $k_m$ denoting the maximum local selection range and $n_t$ denoting the number of remaining available nodes at step $t$. This approach reduces the decision space and accelerates the decision-making process (see Table 5). Formally, we denote the features of nodes $\pi_{t-1}$ and $\pi_1$ and the candidate set $K_{set}$ as $E_{\pi_{t-1}} \in \mathbb{R}^{1 \times h}$, $E_{\pi_1} \in \mathbb{R}^{1 \times h}$, and $E_{K_{set}} \in \mathbb{R}^{k \times h}$, respectively. We concatenate these features to form the decoder's input at MDP step $t$ as follows:

$$D = (E_{\pi_{t-1}}, E_{K_{set}}, \ldots, E_{\pi_1}), D \in \mathbb{R}^{(k+2) \times h}. \quad (10)$$

To capture subtle distinctions between the nodes within the local selection range, we employ the attention mechanism used in Zhou et al. (2024) which integrates the distance matrix $A$ among the decoder input nodes (see detailed mechanisms in Appendix A.2). Additionally, to mitigate potential value overflows due to repeated exponential operations, we incorporate RMSNorm (Zhang & Sennrich, 2019) into the attention mechanism. The time and space complexity of the attention mechanism in our decoder is $\mathcal{O}(k_m^2 h)$ and $\mathcal{O}(k_m^2 + k_m h)$, respectively.

After refining the local node features through multiple attention layers, we compute the probability distribution of nodes in candidate set $K_{set}$ being selected at MDP step $t$ as follows:

$$p_\theta(a_t) = \text{Softmax}\left(D_{x_i} W \odot \begin{cases} 1, & \text{if } x_i \in K_{set}, \\ -\infty, & \text{otherwise,} \end{cases}\right), p_\theta(a_t) \in \mathbb{R}^k, \tag{11}$$

where $D_{x_i}$ denotes the features of node $x_i$ and $\odot$ denotes the element-wise multiplication.

## 4.2 TRAINING STRATEGY OF GELD

Existing neural TSP solvers typically rely on Supervised Learning (SL) (Luo et al., 2023; Drakulic et al., 2023), Reinforcement Learning (RL) (Gao et al., 2024; Fang et al., 2024), or Self-Improvement Learning (SIL) (Luo et al., 2024; Pirnay & Grimm, 2024) for model training. We choose not to use RL due to its requirement of generating a complete solution before calibrating the reward, which normally requires a large amount of computational resources.

Inspired by recent advancements in fine-tuning large models (Han et al., 2024), we propose a two-stage training approach. The first stage involves SL training on small-scale instances, followed by SIL training on larger instances. For the first stage, we adopt the same SL method used by Drakulic et al. (2023) and Luo et al. (2023) and utilize the publicly available training dataset contributed by Luo et al. (2023) to ensure fair comparisons in all relevant experiments.

However, the experimental results reveal that models (e.g., GD (Pirnay & Grimm, 2024)) trained on small-scale TSPs exhibit limited generalization capacity on larger-scale TSPs (see Table 1). We hypothesize that this limitation arises because NN-based models typically map inputs to outputs in a fixed manner. When the node distribution in the test data significantly differs from that in the training data, the model struggles to generalize effectively. In this work, we expand the training data size in the second stage to mitigate the model's reduced effectiveness in solving larger instances. We introduce the mechanisms of each training stage as follows (see Appendix A.3 for more details).

**SL Training on Small-scale TSPs.** We define TSPs with fewer than $k_m$ (i.e., the maximum local selection range) nodes as small-scale TSPs. For a TSP-$n$ training instance $s$, we employ the cross-entropy function to maximize the probability of selecting the optimal action at each step as follows:

$$\mathcal{L}(\theta|s) = -\sum_{i=1}^{n} y_i \log(p_\theta(i)), \tag{12}$$

where $n \leq k_m$, $y_i \in \{0, 1\}$ denotes the ground-truth label, indicating whether node $x_i$ should be selected at the current step, and $p_\theta(i)$ denotes the probability of selecting node $x_i$.

**SIL Training on Large-scale TSPs.** After the first training stage, the model exhibits preliminary generalization capability for solving large-scale TSPs. In the second stage, we enhance the model's generalization ability by applying SIL using larger instances, adhering to a curriculum learning strategy that progressively scales the training instances from the small-scale size $k_m$ to a predefined maximum training size $n_{max}$. Specifically, in each training epoch, we randomly generate a batch of $n_{bs}^t$ training instances and apply both BS and PRC to obtain improved solutions (over those produced by the greedy strategy) as pseudo-labels for training. The epoch concludes when any of the following three conditions is met: 1) The maximum number $t_{max}$ of training iterations per batch is reached; 2) The gap between the greedy and improved solutions falls below a predefined threshold $\epsilon$; or 3) There is no further improvement in solution quality after $t_{imp}$ iterations. Furthermore, to prevent overfitting to large-scale problems and ensure adequate focus on smaller instances, we incorporate $n_{bs}^t$ labeled small-scale TSP-$k_m$ instances into the training set at each epoch[1].

## 5 EXPERIMENTAL RESULTS

In this section, we conduct extensive experiments on both synthetic and real-world datasets to evaluate the performance of GELD as a standalone TSP solver and as a post-processing method. The detailed hyperparameter configurations of GELD are provided in Appendix B.1. The synthetic datasets comprise four distribution patterns (namely uniform, clustered, explosion, and implosion) across five scales (100, 500, 1,000, 5,000, and 10,000 nodes). The real-world datasets comprise two collections: TSPLIB95 and National TSPs. Additionally, we select the four largest TSPs from the World TSP dataset to evaluate GELD's performance on extremely large TSPs. Further details on these

---

[1]The source code of GELD is available online, URL: https://anonymous.4open.science/r/ICLR-13204.

Table 1: Performance comparisons on synthetic TSPs of different sizes and distribution patterns

| Model + Inference | TSP-100 (200) gap(%)↓ | time↓, $n_{bs}$↑ | TSP-500 (200) gap(%)↓ | time↓, $n_{bs}$↑ | TSP-1000 (200) gap(%)↓ | time↓, $n_{bs}$↑ | TSP-5000 (20) gap(%)↓ | time↓, $n_{bs}$↑ | TSP-10000 (20) gap(%)↓ | time↓, $n_{bs}$↑ | Average gap(%)↓ |
|---|---|---|---|---|---|---|---|---|---|---|---|
| **uniform** | | | | | | | | | | | |
| (Near-)Optimality | - | 2.7m, 1 | - | 3.7h, 1 | - | 15.2h, 1 | - | 1.7h, 1 | - | 1.3d, 1 | - |
| Omni-TSP (ICML'23) + G* | 2.22 | **0.3s**, 200 | 7.80 | 9.6s, 200 | 19.56 | 1.2m, 100 | 49.43 | 16.1m, 5 | 61.39 | 2.0h, 1 | 28.09 |
| LEHD (NeurIPS'23) + G | 0.67 | 0.7s, 200 | 1.58 | 16.2s, 200 | 2.76 | 1.8m, 100 | 15.80 | 18.2m, 5 | 24.10 | 2.3h, 1 | 8.96 |
| BQ (NeurIPS'23) + G | 5.37 | 1.5s, 200 | 3.86 | 1.3m, 200 | 3.82 | 9.3m, 100 | 12.68 | 1.9h, 5 | 18.74 | 13.5h, 1 | 8.85 |
| ELG (IJCAI'24) + G* | 0.58 | 0.5s, 200 | 8.80 | 4.2s, 200 | 12.22 | 15.6s, 200 | 18.84 | 40.5s, 5 | 18.32 | 3.7m, 2 | 11.77 |
| INViT-3V (ICML'24) + G† | 1.47 | 15.2s, 200 | 4.26 | 1.5m, 200 | 4.96 | 3.1m, 200 | 6.60 | 4.4m, 20 | 4.80 | 6.5m, 20 | 4.42 |
| GD (TMLR'24) + G | 0.72 | 3.1s, 200 | 2.25 | 36.4s, 200 | 4.26 | 3.2m, 200 | 60.26 | 26.7m, 20 | 198.65 | 3.4h, 4 | 53.22 |
| UDC§ (NeurIPS'24) + G* | 0.40 | 8.7s, 200 | 2.15 | 28.5s, 200 | 2.06 | 57.2s, 100 | 6.99 | 29.7s, 20 | 8.73 | 2.4m, 1 | 4.07 |
| GELD (Ours) + G | 1.11 | 0.6s, 200 | 2.39 | **1.8s**, 200 | 2.94 | **3.6s**, 200 | 7.62 | **10.8s**, 20 | 9.33 | **21.6s**, 20 | 4.68 |
| GELD (Ours) + BS(16) | 0.12 | 4.2s, 200 | 0.99 | 32.4s, 200 | 1.30 | 1.1m, 200 | 5.32 | 36.6s, 20 | 6.71 | 1.2m, 20 | 2.89 |
| GELD (Ours) + PRC(100) | 0.81 | 1.8s, 200 | 1.90 | 9.0s, 200 | 1.68 | 18.6s, 200 | 4.66 | 17.4s, 20 | 5.75 | 39.6s, 20 | 2.96 |
| GELD (Ours) + BS(16) + PRC(100) | 0.09 | 5.4s, 200 | 0.83 | 36.9s, 200 | 0.85 | 1.4m, 200 | 3.39 | 44.4s, 20 | 4.19 | 1.6m, 20 | 1.87 |
| GELD (Ours) + BS(16) + PRC(1000) | **0.06** | 19.2s, 200 | **0.52** | 1.6m, 200 | **0.58** | 3.7m, 200 | **2.77** | 1.8m, 20 | **2.38** | 3.9m, 20 | **1.26** |
| **clustered** | | | | | | | | | | | |
| (Near-)Optimality | - | 3.1m, 1 | - | 4.1h, 1 | - | 16.1h, 1 | - | 3.0h, 1 | - | 1.5d, 1 | - |
| Omni-TSP (ICML'23) + G* | 2.37 | **0.3s**, 200 | 9.82 | 9.6s, 200 | 21.20 | 1.2m, 100 | 54.49 | 16.1m, 5 | 71.60 | 2.0h, 1 | 26.56 |
| LEHD (NeurIPS'23) + G | 1.43 | 0.7s, 200 | 4.60 | 16.2s, 200 | 8.56 | 1.8m, 100 | 23.46 | 18.2m, 5 | 35.33 | 2.3h, 1 | 12.30 |
| BQ (NeurIPS'23) + G | 5.33 | 1.5s, 200 | 6.66 | 1.3m, 200 | 9.43 | 9.3m, 100 | 27.65 | 1.9h, 5 | 41.80 | 13.5h, 1 | 15.21 |
| ELG (IJCAI'24) + G* | 2.67 | 0.5s, 200 | 11.31 | 4.2s, 200 | 15.27 | 15.6s, 200 | 25.73 | 40.5s, 5 | 31.01 | 3.7m, 2 | 14.34 |
| INViT-3V (ICML'24) + G† | 2.29 | 15.2s, 200 | 5.21 | 1.5m, 200 | 6.03 | 3.1m, 200 | 7.17 | 4.4m, 20 | 6.31 | 6.5m, 20 | 4.49 |
| GD (TMLR'24) + G | 2.29 | 3.1s, 200 | 6.87 | 36.4s, 200 | 25.26 | 3.2m, 200 | 329.10 | 26.7m, 20 | 627.83 | 3.4h, 4 | 198.41 |
| UDC§ (NeurIPS'24) + G* | 2.54 | 8.7s, 200 | 5.89 | 28.6s, 100 | 8.26 | 57.2s, 100 | 15.19 | 29.5s, 20 | 15.41 | 2.4m, 1 | 9.46 |
| GELD (Ours) + G | 3.28 | 0.6s, 200 | 4.41 | **1.8s**, 200 | 5.93 | **3.6s**, 200 | 11.62 | **10.8s**, 20 | 12.53 | **21.6s**, 20 | 7.55 |
| GELD (Ours) + BS(16) | 1.32 | 4.2s, 200 | 3.14 | 32.4s, 200 | 4.82 | 1.1m, 200 | 8.92 | 36.6s, 20 | 9.61 | 1.2m, 20 | 5.56 |
| GELD (Ours) + PRC(100) | 2.24 | 1.8s, 200 | 3.27 | 9.0s, 200 | 3.35 | 18.6s, 200 | 6.59 | 17.4s, 20 | 7.87 | 39.6s, 20 | 4.66 |
| GELD (Ours) + BS(16) + PRC(100) | 0.92 | 5.4s, 200 | 2.50 | 36.9s, 200 | 2.94 | 1.4m, 200 | 5.16 | 44.4s, 20 | 5.92 | 1.6m, 20 | 3.54 |
| GELD (Ours) + BS(16) + PRC(1000) | **0.46** | 19.2s, 200 | **1.23** | 1.6m, 200 | **2.24** | 3.7m, 200 | **4.27** | 1.8m, 20 | **3.44** | 3.9m, 20 | **2.33** |
| **explosion** | | | | | | | | | | | |
| (Near-)Optimality | - | 2.7m, 1 | - | 3.8h, 1 | - | 15.6h, 1 | - | 1.7h, 1 | - | 1.3d, 1 | - |
| Omni-TSP (ICML'23) + G* | 2.05 | **0.3s**, 200 | 9.25 | 9.6s, 200 | 19.95 | 1.2m, 100 | 51.28 | 16.1m, 5 | 65.37 | 2.0h, 1 | 24.69 |
| LEHD (NeurIPS'23) + G | 0.63 | 0.7s, 200 | 2.65 | 16.2s, 200 | 5.76 | 1.8m, 100 | 21.07 | 18.2m, 5 | 30.55 | 2.3h, 1 | 10.12 |
| BQ (NeurIPS'23) + G | 5.97 | 1.5s, 200 | 4.88 | 1.3m, 200 | 7.11 | 9.3m, 100 | 29.39 | 1.9h, 5 | 51.54 | 13.5h, 1 | 16.41 |
| ELG (IJCAI'24) + G* | 0.87 | 0.5s, 200 | 9.27 | 4.2s, 200 | 13.67 | 15.6s, 200 | 22.79 | 40.5s, 5 | 23.46 | 3.7m, 2 | 11.68 |
| INViT-3V (ICML'24) + G† | 1.62 | 15.2s, 200 | 5.54 | 1.5m, 200 | 7.32 | 3.1m, 200 | 9.92 | 4.4m, 20 | 7.85 | 6.5m, 20 | 5.37 |
| GD (TMLR'24) + G | 0.68 | 3.1s, 200 | 3.32 | 36.4s, 200 | 12.33 | 3.2m, 200 | 271.55 | 26.7m, 20 | 682.40 | 3.4h, 4 | 194.07 |
| UDC§ (NeurIPS'24) + G* | 0.66 | 8.6s, 200 | 4.60 | 28.6s, 200 | 6.96 | 57.2s, 100 | 16.15 | 29.5s, 20 | 17.44 | 2.4m, 1 | 9.16 |
| GELD (Ours) + G | 1.67 | 0.6s, 200 | 3.79 | **1.8s**, 200 | 5.40 | **3.6s**, 200 | 12.13 | **10.8s**, 20 | 14.27 | **21.6s**, 20 | 7.45 |
| GELD (Ours) + BS(16) | 0.41 | 4.2s, 200 | 2.39 | 32.4s, 200 | 3.62 | 1.1m, 200 | 9.52 | 36.6s, 20 | 11.13 | 1.2m, 20 | 5.41 |
| GELD (Ours) + PRC(100) | 0.96 | 1.8s, 200 | 2.76 | 9.0s, 200 | 2.90 | 18.6s, 200 | 7.13 | 17.4s, 20 | 9.28 | 39.6s, 20 | 4.61 |
| GELD (Ours) + BS(16) + PRC(100) | 0.27 | 5.4s, 200 | 1.74 | 36.9s, 200 | 2.23 | 1.4m, 200 | 5.86 | 44.4s, 20 | 7.45 | 1.6m, 20 | 3.51 |
| GELD (Ours) + BS(16) + PRC(1000) | **0.18** | 19.2s, 200 | **0.95** | 1.6m, 200 | **1.52** | 3.7m, 200 | **4.55** | 1.8m, 20 | **4.70** | 3.9m, 20 | **2.39** |
| **implosion** | | | | | | | | | | | |
| (Near-)Optimality | - | 2.6m, 1 | - | 3.6h, 1 | - | 15.2h, 1 | - | 2.2h, 1 | - | 1.3d, 1 | - |
| Omni-TSP (ICML'23) + G* | 2.04 | **0.3s**, 200 | 8.63 | 9.6s, 200 | 19.18 | 1.2m, 100 | 50.37 | 16.1m, 5 | 62.58 | 2.0h, 1 | 23.83 |
| LEHD (NeurIPS'23) + G | 1.13 | 0.7s, 200 | 2.57 | 16.2s, 200 | 4.10 | 1.8m, 100 | 17.48 | 18.2m, 5 | 26.46 | 2.3h, 1 | 8.62 |
| BQ (NeurIPS'23) + G | 5.44 | 1.5s, 200 | 4.84 | 1.3m, 200 | 5.22 | 9.3m, 100 | 16.42 | 1.9h, 5 | 25.23 | 13.5h, 1 | 9.56 |
| ELG (IJCAI'24) + G* | 0.91 | 0.5s, 200 | 8.44 | 4.2s, 200 | 12.40 | 15.6s, 200 | 18.95 | 40.5s, 5 | 18.73 | 3.7m, 2 | 9.89 |
| INViT-3V (ICML'24) + G† | 1.79 | 15.2s, 200 | 4.84 | 1.5m, 200 | 5.64 | 3.1m, 200 | 6.85 | 4.4m, 20 | 5.41 | 6.5m, 20 | 4.07 |
| GD (TMLR'24) + G | 1.45 | 3.1s, 200 | 4.29 | 36.4s, 200 | 8.68 | 3.2m, 200 | 100.05 | 26.7m, 20 | 259.46 | 3.4h, 4 | 74.74 |
| UDC§ (NeurIPS'24) + G* | 0.54 | 8.7s, 200 | 3.29 | 28.7s, 200 | 3.74 | 57.2s, 100 | 7.74 | 29.5s, 20 | 10.04 | 2.4m, 1 | 5.07 |
| GELD (Ours) + G | 2.23 | 0.6s, 200 | 4.71 | **1.8s**, 200 | 4.98 | **3.6s**, 200 | 9.23 | **10.8s**, 20 | 10.02 | **21.6s**, 20 | 6.25 |
| GELD (Ours) + BS(16) | 0.83 | 4.2s, 200 | 4.06 | 32.4s, 200 | 4.07 | 1.1m, 200 | 6.13 | 36.6s, 20 | 7.45 | 1.2m, 20 | 4.52 |
| GELD (Ours) + PRC(100) | 1.55 | 1.8s, 200 | 3.54 | 9.0s, 200 | 2.93 | 18.6s, 200 | 5.68 | 17.4s, 20 | 6.19 | 39.6s, 20 | 3.96 |
| GELD (Ours) + BS(16) + PRC(100) | 0.53 | 5.4s, 200 | 3.22 | 36.9s, 200 | 2.54 | 1.4m, 200 | 4.06 | 44.4s, 20 | 4.71 | 1.6m, 20 | 3.02 |
| GELD (Ours) + BS(16) + PRC(1000) | **0.22** | 19.2s, 200 | **1.29** | 1.6m, 200 | **1.64** | 3.7m, 200 | **3.14** | 1.8m, 20 | **2.81** | 3.9m, 20 | **1.84** |

Symbols "G", "G*", "G†", "BS($i$)", and "PRC($j$)" denote the greedy strategy, greedy multiple rollouts (Kwon et al., 2020), greedy multiple rollouts with data augment technique (Fang et al., 2024), BS with a width of $i$, and PRC with $j$ iterations, respectively. The number in parentheses following "TSP-$n$" indicates the total number of TSP-$n$ test instances. Symbol "§" indicates the model adopts a D&C strategy.

datasets are presented in Appendix B.2. For performance comparisons, we select seven SOTA models as baselines, with their settings outlined in Appendix B.3. For all baseline models and GELD, we report their average gap to the (near-)optimal solutions, inference time, and parallel processing capability on the test datasets (see Appendix B.4 for more details).

## 5.1 PERFORMANCE ANALYSIS OF GELD ON SYNTHETIC AND REAL-WORLD DATASETS

We analyze GELD's performance on synthetic and real-world datasets, respectively.

**Synthetic Datasets.** We present the performance comparison of GELD against baseline models on synthetic datasets in Table 1. The results indicate that all models, including ours, exhibit performance degradation when generalizing to TSPs of varying scales, underscoring the critical need for investigating model generalization. Despite the overall trend of declining performance, our proposed GELD, when paired with the greedy strategy, achieves solution quality on-par with the SOTA INViT-3V model, which employs greedy multiple rollouts and data augment techniques. Moreover, GELD offers a significant advantage in inference speed, consistently outperforming other models across different scales, except for TSP-100. This can be attributed to the efficient, low time complexity backbone architecture of our model. Furthermore, when integrated with BS and PRC, GELD achieves the highest solution quality across all scales and patterns. This superior performance arises from its design, which incorporates diversified model inputs to enhance the effectiveness of RC. Additionally, GELD's capability to process all $n_{bs}$ test instances simultaneously across all scales makes it particularly well-suited for practical applications with limited computing resources.

**Real-world Datasets.** We present the performance comparison of GELD against baselines on real-world datasets in Table 2. For clarity, the experimental results are grouped by the scale, with detailed

Table 2: Performance comparisons on real-world TSPLIB95 and National TSP instances

| TSP-{set} | | <101 | 101-500 | 501-1000 | 1001-5000 | 5001-10000 | >10000 | (Total) gap↓, time↓ |
|---|---|---|---|---|---|---|---|---|
| | Total number of instances | 12 | 30 | 6 | 22 | 2 | 5 | 77 |
| TSPLIB95 | Omni-TSP (ICML'23) + G* | 6.87% | 8.79% | 19.59% | 32.31% | 63.28% | OOM | (72) 18.07%, 3.8s |
| | LEHD (NeurIPS'23) + G | 0.61% | 2.96% | 4.05% | 11.27% | 24.14% | (3) 50.21% | (75) 7.56%, 47.0s |
| | BQ (NeurIPS'23) + G | 8.64% | 8.40% | 8.08% | 13.33% | 27.37% | (1) 45.21% | (73) 10.92%, 1.4m |
| | ELG (IJCAI'24) + G* | 1.56% | 4.55% | 9.25% | 12.61% | 17.31% | OOM | (72) 7.25%, 6.1s |
| | INViT-3V (ICML'24) + G† | 1.15% | 3.38% | 6.33% | 7.47% | 9.34% | 7.57% | 4.86%, 26.2s |
| | GD (TMLR'24) + G | 1.78% | 4.29% | 8.53% | 52.17% | 325.62% | 991.24% | 90.34%, 2.7m |
| | UDC§ (NeurIPS'24) + G* | (6) 0.19% | 2.18% | 10.58% | 13.00% | 26.26% | (1) 23.37% | (67) 7.34%, 5.6s |
| | GELD (Ours) + G | 0.89% | 4.92% | 4.43% | 8.91% | 11.76% | 15.90% | 6.28%, 3.8s |
| | GELD (Ours) + BOTH | **0.26%** | **1.56%** | **1.92%** | **3.44%** | **7.09%** | **5.96%** | **2.35%**, 27.6s |
| | Total number of instances | 2 | 1 | 3 | 4 | 9 | 8 | 27 |
| National TSPs | Omni-TSP (ICML'23) + G* | 2.63% | 10.44% | 17.88% | 71.65% | 83.24% | (1) 71.67% | (20) 58.83%, 2.3m |
| | LEHD (NeurIPS'23) + G | 0.12% | 27.15% | 44.20% | 56.58% | 93.92% | (1) 98.52% | (20) 66.51%, 2.8m |
| | BQ (NeurIPS'23) + G | 24.29% | 12.18% | 10.25% | 40.55% | 94.96% | (1) 55.65% | (20) 58.20%, 14.8m |
| | ELG (IJCAI'24) + G* | 2.28% | 7.06% | 12.55% | 34.93% | 48.95% | (1) 22.44% | (20) 32.60%, 3.7m |
| | INViT-3V (ICML'24) + G† | 0.03% | 2.88% | 5.63% | 10.17% | 11.17% | (7) 9.48% | (26) 8.75%, 3.4m |
| | GD (TMLR'24) + G | 3.51% | 236.40% | 921.61% | 2093.71% | 3868.87% | (4) 5236.23% | (23) 2919.47%, 8.9m |
| | UDC§ (NeurIPS'24) + G* | - | 0.58% | 10.04% | 18.18% | 25.44% | (1) 18.41% | (18) 19.49%, 6.3s |
| | GELD (Ours) + G | 0.41% | 0.53% | 5.10% | 14.80% | 17.99% | 18.80% | 14.39%, 23.4s |
| | GELD (Ours) + BOTH | **0.02%** | **0.02%** | **2.12%** | **6.97%** | **7.66%** | **8.21%** | **6.26%**, 1.4m |

For each model, we report the average gap and inference time for the instances it successfully solves within a given set. We use "BOTH" to denote the operation of BS(16) + PRC(1000) for brevity. Symbol "OOM" (Out of Memory) is used to indicate cases where the model fails to solve all instances in the set due to the GPU memory constraint. Symbol "$(i)$" denotes the number of instances the model successfully solves in this set. The absence of these two symbols indicates that the model can solve all instances in the set. Moreover, UDC fails to solve instances with sizes smaller than 100 nodes due to unknown errors.

Table 3: Performance of baselines on National TSPs using GELD as a post-processing method

| TSP-{set} | <101 | 101-500 | 501-1000 | 1001-5000 | 5001-10000 | >10000 | (Total) gap↓, time↓ | Gain↑ |
|---|---|---|---|---|---|---|---|---|
| Total number | 2 | 1 | 3 | 4 | 9 | 8 | 27 | |
| Omni-TSP + GELD | 0.02% | 0.67% | 1.61% | 5.49% | 7.13% | (1) 5.40% | (20) 4.85%, +26.6s | **91.76%** |
| LEHD + GELD | 0.02% | 7.12% | 3.24% | 8.40% | 9.30% | (1) 11.64% | (20) 7.29%, +26.6s | 89.04% |
| BQ + GELD | 0.02% | 4.52% | 3.48% | 7.01% | 9.59% | (1) 8.81% | (20) 6.91%, +26.6s | 88.13% |
| ELG + GELD | 2.15% | 0.67% | 2.56% | 8.78% | 17.00% | (1) 7.99% | (20) 10.44%, +26.6s | 67.98% |
| INViT-3V + GELD | 0.02% | 0.68% | 2.24% | 4.57% | 5.35% | (7) 4.74% | (26) 4.12%, +34.0s | 52.91% |
| GD + GELD | 0.29% | 4.52% | 22.66% | 65.78% | 131.14% | (4) 140.89% | (23) 90.43%, +28.3s | **96.90%** |
| UDC + GELD | - | 0.58% | 7.59% | 11.30% | 16.45% | (1) 11.19% | (18) 12.66%, +26.2s | 35.05% |
| Random Insertion | 8.77% | 11.54% | 11.53% | 12.49% | 13.48% | 13.34% | 12.66%, 1.1s | **78.59%** |
| + GELD | 0.02% | 2.35% | 1.75% | 3.14% | 3.29% | 2.90% | **2.71%**, +36.8s | |

Gain is calculated as 1-(the result of baseline with GELD)/(the result of baseline without GELD).

performance presented in Appendix B.5. The results demonstrate that GELD consistently outperforms baseline models across all sets of TSP instances in terms of both solution quality and inference speed. Additionally, due to the GPU memory constraint (24GB), all baseline models are unable to solve certain large-scale TSP instances, whereas our model successfully solves all instances. This advantage is attributed to the low space complexity of our model's backbone architecture, again underscoring its suitability for practical applications with limited computing resources.

## 5.2 PERFORMANCE OF GELD AS A POST-PROCESSING TECHNIQUE FOR BASELINES

We apply GELD in combination with PRC(1000) to assess its effectiveness as a post-processing method for improving the solution quality of baseline models. Because the baseline models struggle with certain large-scale instances (e.g., CH71009 with 71,009 nodes), we introduce a simple and generic heuristic—Random Insertion—as an additional baseline. Random Insertion greedily selects the insertion point for each node, minimizing the insertion cost. We use the National TSPs dataset as the benchmark and apply GELD to reconstruct the solution generated by these baselines.

The results, as presented in Table 3, demonstrate that our model significantly improves the solution quality by at least **35%** with an affordable increase in computing time, thereby highlighting the efficacy of GELD as a post-processing method. Moreover, the successful integration with Random Insertion, characterized by low latency and high solution quality, suggests that combining GELD with heuristic algorithms is a promising approach for efficiently solving large-scale TSPs.

To further demonstrate the effectiveness of combining GELD with heuristic algorithms, we conduct additional experiments on the four extremely large TSPs, with sizes ranging from 104,815 to

Table 4: Performance of GELD on extremely large TSP instances

| Instances | | sra104815 | ara238025 | lra498378 | lrb744710 |
|---|---|---|---|---|---|
| Random Insertion | gap | 21.26% | 20.65% | 18.94% | 21.08% |
| | time | 52.2s | 5.39m | 30.1m | 1.69h |
| +GELD | gap (gain) | 9.67% (54.66%) | 9.25% (55.21%) | 6.58% (65.26%) | 8.97% (57.45%) |
| | time | +2.7m | +5.9m | +12.9m | +19.7m |

Table 5: Ablation studies on synthetic TSP instances of the uniform distribution

| Model + Inference | | TSP-100 (200) gap(%)↓ | time↓, $n_{bs}$ ↑ | TSP-500 (200) gap(%)↓ | time↓, $n_{bs}$ ↑ | TSP-1000 (200) gap(%)↓ | time↓, $n_{bs}$ ↑ | TSP-5000 (20) gap(%)↓ | time↓, $n_{bs}$ ↑ | TSP-10000 (20) gap(%)↓ | time↓, $n_{bs}$ ↑ |
|---|---|---|---|---|---|---|---|---|---|---|---|
| w/o RALA | G | 1.12 | 0.8s, 200 | 2.61 | 2.0s, 200 | 3.63 | 4.1s, 200 | 11.67 | 45.2s, 5 | 12.48 | 3.7m, 2 |
| | BOTH | 0.05 | 20.1s, 200 | 0.48 | 1.7m, 200 | 0.64 | 3.6m, 200 | 4.18 | 3.9m, 5 | 4.04 | 27.3m, 1 |
| | - Norm | 0.05 | 20.1s, 200 | 0.50 | 1.7m, 200 | 0.72 | 3.6m, 200 | 5.25 | 3.9m, 5 | 5.76 | 27.3m, 1 |
| w/o second stage training | G | 0.86 | 0.6s, 200 | 3.28 | 1.8s, 200 | 4.17 | 3.6s, 200 | 13.61 | 10.8s, 20 | 15.21 | 21.6s, 20 |
| | BOTH | 0.05 | 19.2s, 200 | 0.69 | 1.6m, 200 | 1.14 | 3.7m, 200 | 3.73 | 1.8m, 20 | 3.10 | 3.9m, 20 |
| | - Norm | 0.05 | 19.2s, 200 | 0.74 | 1.6m, 200 | 1.39 | 3.7m, 200 | 5.50 | 1.8m, 20 | 5.62 | 3.9m, 20 |
| w/o global view | G | 1.33 | 0.5s, 200 | 3.03 | 1.5s, 200 | 3.79 | 3.2s, 200 | 10.14 | 9.9s, 20 | 11.13 | 20.2s, 20 |
| | BOTH | 0.06 | 18.5s, 200 | 0.54 | 1.6m, 200 | 0.65 | 3.5m, 200 | 3.08 | 1.8m, 20 | 3.41 | 3.8m, 20 |
| | - Norm | 0.06 | 18.5s, 200 | 0.54 | 1.6m, 200 | 0.67 | 3.5m, 200 | 3.84 | 1.8m, 20 | 4.78 | 3.8m, 20 |
| w/o local view | G | 1.32 | 0.6s, 200 | 2.13 | 6.1s, 200 | 2.51 | 33.6s, 200 | 4.82 | 4.1m, 20 | 5.57 | 30.9m, 5 |
| | BOTH | 0.08 | 19.2s, 200 | 0.44 | 3.0m, 100 | 0.42 | 12.6m, 40 | 1.92 | 1.4h, 1 | OOM | |
| | - Norm | 0.09 | 19.2s, 200 | 0.45 | 3.0m, 100 | 0.47 | 12.6m, 40 | 2.28 | 1.4h, 1 | | |
| GELD | G | 1.11 | 0.6s, 200 | 2.39 | 1.8s, 200 | 2.94 | 3.6s, 200 | 7.62 | 10.8s, 20 | 9.33 | 21.6s, 20 |
| | BOTH | 0.06 | 19.2s, 200 | 0.52 | 1.6m, 200 | 0.58 | 3.7m, 200 | 2.77 | 1.8m, 20 | 2.38 | 3.9m, 20 |
| | - Norm | 0.06 | 19.2s, 200 | 0.53 | 1.6m, 200 | 0.61 | 3.7m, 200 | 3.29 | 1.8m, 20 | 3.64 | 3.9m, 20 |

Symbol "- Norm" denotes without the normalization operation during the RC process.

744,710 nodes. As shown in Table 4, GELD efficiently solves these extremely large TSPs. To the best of our knowledge, **our proposed approach is the first neural model capable of solving TSPs with up to 744,710 nodes without relying on D&C strategies**.

### 5.3 ABLATION STUDIES ON GELD DESIGN CHOICES

We conduct extensive ablation studies to assess the effectiveness of the key design choices in GELD, by investigating the following five aspects: 1) The efficacy of RALA; 2) The impact of the second-stage training; 3) The benefit of the global view in GE; 4) The importance of the local view in LD; and 5) The effectiveness of diversifying model inputs.

We present the ablation study results in Table 5. Firstly, while GELD with the standard attention mechanism performs comparably to GELD with RALA on small-scale instances (TSP-{100, 500}), it experiences a performance degradation (especially in inference speed and parallel processing capability) on large-scale instances (TSP-{5000, 10000}). This finding demonstrates that RALA is critical for enabling GELD to solve TSPs in a short time period and scale effectively to larger instances. Secondly, incorporating the second-stage training leads to a 39.1% improvement in solution quality compared to only applying the first-stage training, underscoring the importance of the two-stage training strategy. Notably, even without the second-stage training, GELD achieves an average gap of 1.74%, outperforming all seven baseline models (see Table 1). Thirdly, integrating the global view into GELD improves the average gap by 31.6% when compared to using a local view only (i.e., removing the global information transfer module from GE), demonstrating the benefit of exploiting global information. Fourthly, while extending LD's local view to a global view (i.e., considering all available nodes as candidates instead of set $K_{set}$) enhances solution quality, it significantly hampers inference speed and parallel processing capability, particularly in large-scale instances (TSP-10000). These results highlight the effectiveness of the local view in enabling GELD to efficiently solve TSPs of varying sizes. Last but not least, removing the normalization operation in the RC process deteriorates model performance in all aspects, illustrating the importance of diversifying model inputs.

## 6 CONCLUSION

In this study, we positively answer the proposed research question with ample experimental results as supporting evidence. Specifically, we introduce GELD, which effectively solves TSPs of all sizes while capable of exchanging affordable computing time for significantly improved solution quality. We believe the proposed *broad global assessment and refined local selection* framework will offer valuable insights towards solving other COPs. Going forward, we plan to extend the capability of GELD to solve more complex COPs, such as the capacitated vehicle routing problem and JSSP.

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

# A APPENDIX OF METHOD

## A.1 DETAILED RE-CONSTRUCTION PROCESS

This section presents the detailed RC process employed in our study, comprising two main steps. Firstly, after obtaining the initial solutions, denoted as $\pi = \{\pi_1, \pi_2, ..., \pi_n\}$, RC randomly selects a starting index $i$ and a sub-solution length $j$ to form a sub-solution $\{\pi_i, \pi_{i+1}, ..., \pi_{i+j}\}$, where $i \in \{1, ..., n\}$ and $j > 2$. This condition ensures that the sub-solution length is sufficient to impact the outcome, as sub-problems smaller than size 4 do not alter the sub-solutions during the RC process (i.e., there must be at least two nodes in the candidate set $K_{set}$ for selection). Since the TSP solution $\pi$ forms a cyclic sequence, i.e., $\pi_{n+i} = \pi_i$, RC adapts the sampling direction based on the iteration count, alternating between clockwise ($\pi_i, \pi_{i+1}, ..., \pi_{i+j}$) and counterclockwise ($\pi_i, \pi_{i-1}, ..., \pi_{i-j}$). To further enhance model input diversity, the solution sequence is shifted by a randomly selected offset $n_\epsilon$ from $\{1, ..., n\}$. In the second step, RC reintegrates the selected node features into the model. To introduce additional model input diversity, we randomly apply one of the $\times 8$ data augmentation techniques proposed by Kwon et al. (2020), such as rotating the TSP topology by 90 degrees. The model then generates new sub-solutions using a greedy strategy. If these newly generated sub-solutions outperform the existing ones, they replace the current sub-solutions.

## A.2 DETAILED COMPUTATION PROCESS OF ATTENTION MECHANISM USED IN DECODER

This section outlines a detailed computational process of the attention mechanism employed in the decoder, as introduced by (Zhou et al., 2024). Specifically, given the query $Q = DW$, key $K = DW$, and value $V = DW$, where $Q, K, V \in \mathbb{R}^{(k+2) \times h}$, the updated embedding is computed as follows:

$$D = \text{Sigmoid}(Q) \odot \frac{\exp(A_{tmp})(\exp(K) \odot V)}{\exp(A_{tmp}) \exp(K)}, D \in \mathbb{R}^{(k+2) \times h}, \tag{13}$$

$$A_{tmp} = \alpha \cdot \log_2(k+2) \cdot A, A_{tmp} \in \mathbb{R}^{(k+2) \times (k+2)}, \tag{14}$$

where $\alpha$ denotes a learnable parameter.

## A.3 TRAINING ALGORITHM OF GELD

We adopt the SL method used in Drakulic et al. (2023); Luo et al. (2023), which enhances the diversity of training data by focusing on partial optimal solutions. Given a solution $\pi = \{\pi_1, \pi_2, ..., \pi_n\}$—either ground-truth or pseudo labels—we randomly select a partial solution for model training, e.g., $\{\pi_i, \pi_{i+1}, ..., \pi_{i+j}\}$, where $j > 2$. Furthermore, we present the overall two-stage training strategy of GELD in Algorithm 1.

# B APPENDIX OF EXPERIMENTS

## B.1 HYPERPARAMETER CONFIGURATION

We follow the convention and focus on the $d = 2$-dimensional TSP (Kwon et al., 2020). Our proposed GELD comprises 1 (*broad*) global-view encoder layer and 6 (*refined*) local-view decoder layers, each with a hidden dimension of $h = 128$ and 8 attention heads, following Luo et al. (2023). To balance performance and computational complexity, we set the numbers of rows and columns to $m_r = 3$ and $m_c = 3$, respectively, resulting in $m = m_r \cdot m_c = 3 \times 3 = 9$ regions. For fair comparisons with SL-trained models (Drakulic et al., 2023; Luo et al., 2023), we adhere to the same training scale, with the small-scale size set to $k_m = 100$. While increasing the maximum training size $n_{max}$ intuitively improves model generalization, it also increases computational costs. To strike a balance, we set $n_{max}$ to 1000. In the first training stage, we utilize the publicly available training dataset $data_s$ from Luo et al. (2023) for fair comparisons. The learning rate is set to 1e–4 with a decay rate of 0.97. In the second training stage, the training termination hyperparameters are set to $t_{max} = 5, \epsilon = $1e–3, and $t_{imp} = 3$. The learning rate is adjusted to 1e–5, and the training batch size $n_{bs}^t$ is set to 64. The width of BS and iteration number of PRC are set to 16 and 1,000, respectively. All training instances are randomly generated by sampling the node locations based on the uniform

---

**Algorithm 1** Two-stage training strategy of GELD.

---

**Input:** the small-scale TSP-$k_m$ dataset $data_s$, training batch size $n_{bs}^t$, maximum training size $n_{max}$, epoch numbers $n_{e1}$ and $n_{e2}$ for the first stage and the second stage, respectively, training termination hyperparameters $t_{max}, \epsilon$, and $t_{imp}$.

  1: Initialize $\theta$
  2: *The first-stage SL training on small-scale TSPs*
  3: **for** *epoch* in $1, ..., n_{e1}$ **do**
  4:     $data_1, label_1 \leftarrow$ TSP-$n$ instances from $data_s$, where $n \leq k_m$
  5:     $\theta \leftarrow$ **GELD**$(\theta, data_1, label_1)$
  6: **end for**
  7: *The second-stage SIL training on large-scale TSPs*
  8: **for** *epoch* in $1, ..., n_{e2}$ **do**
  9:     $l_{scale} \leftarrow k_m + epoch \cdot (n_{max} - k_m) \mid n_{e2}$
10:     $data_2 \leftarrow$ Randomly generate $n_{bs}^t$ TSP-$l_{scale}$ instances
11:     $len_G, \_ \leftarrow$ Greedy strategy$(\textbf{GELD}, data_2)$
12:     $len_I, solution \leftarrow$ PRC(BS$(\textbf{GELD}, data_2))$
13:     $t_1 \leftarrow 0, t_2 \leftarrow 0$
14:     **while** $t_1 < t_{max}$ and $\frac{len_G}{len_I} - 1 > \epsilon$ and $t_2 < t_{imp}$ **do**
15:         $data_1, label_1 \leftarrow$ Randomly sample $n_{bs}^t$ TSP-$k_m$ instances from $data_s$
16:         $data, label \leftarrow \{data_2, solution\} \cup \{data_1, label_1\}$
17:         $\theta \leftarrow$ **GELD**$(\theta, data, label)$
18:         $len_G, \_ \leftarrow$ Greedy strategy$(\textbf{GELD}, data_2)$
19:         $len_{I_{tmp}}, solution_{tmp} \leftarrow$ PRC(BS$(\textbf{GELD}, data_2))$
20:         **if** $len_{I_{tmp}} < len_I$ **then**
21:             $t_2 \leftarrow 0, len_I \leftarrow len_{I_{tmp}}, solution \leftarrow solution_{tmp}$
22:         **else**
23:             $t_2 \leftarrow t_2 + 1$
24:         **end if**
25:         $t_1 \leftarrow t_1 + 1$
26:     **end while**
27: **end for**

---

distribution pattern. To control the overall training time—approximately 20 hours for the first stage and 31 hours for the second stage—we set the number of epochs $n_{e1} = 50$ and $n_{e2} = 50$ for the first and second stages, respectively. All experiments were conducted on a computer equipped with an Intel(R) Core(TM) i9-12900K CPU and an NVIDIA RTX 4090 GPU (24GB).

## B.2    DATASETS COMPONENT

We conduct a comprehensive evaluation of model performance using both synthetic datasets and widely recognized real-world benchmark datasets.

**Synthetic Datasets.** For the synthetic data, we generate TSP instances of varying sizes and distributions. Specifically, we synthesize 20 subsets of TSP instances, encompassing four distribution patterns (uniform, clustered, explosion, and implosion) across five scales (100, 500, 1,000, 5,000, and 10,000 nodes), following Fang et al. (2024); Bossek et al. (2019). We provide a visualization of TSP-10000 instances for each distribution patterns in Figure 3. The number of instances per subset is determined by the scale, comprising 200 instances for TSP-100, TSP-500, and TSP-1000, and 20 instances for TSP-5000 and TSP-10000.

**Real-world Datasets.** To assess the model's performance in real-world scenarios, we utilize the widely recognized TSPLIB and World TSP datasets as benchmarks. For TSPLIB, we include all symmetric instances from TSPLIB95[2] with nodes represented as Euclidean 2D coordinates, covering 77 instances with sizes ranging from 51 to 18,512 nodes. For World TSP, we include all sym-

---

[2]URL: http://comopt.ifi.uni-heidelberg.de/software/TSPLIB95/

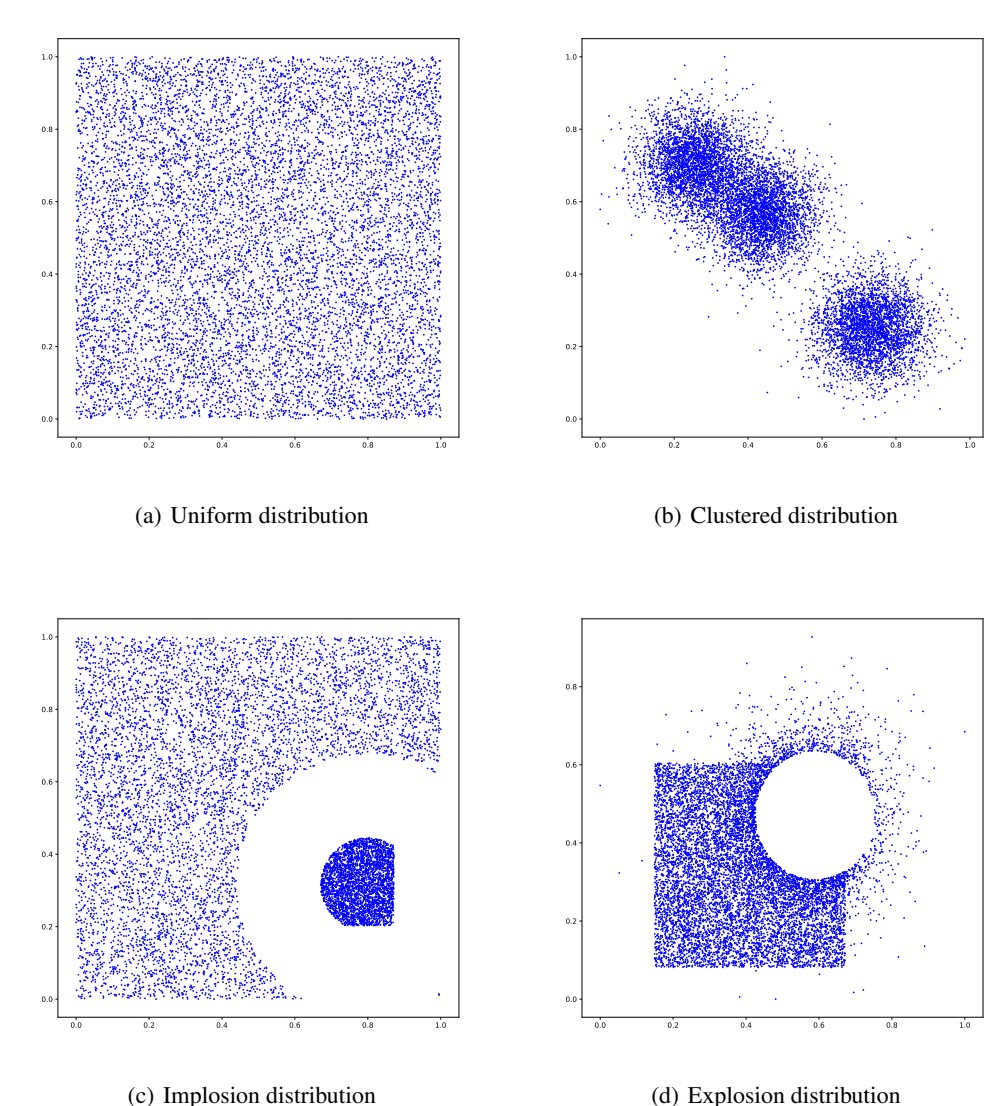

(a) Uniform distribution

(b) Clustered distribution

(c) Implosion distribution

(d) Explosion distribution

Figure 3: Visualization of TSP-10000 instances (synthetic) with four distribution patterns.

metric instances from National TSPs[3], also represented as Euclidean 2D coordinates, comprising 27 instances with sizes ranging from 29 to 71,009 nodes.

**Extremely Large TSP Instances.** To assess the performance of combining GELD with heuristic algorithms, we utilize the four largest TSP instances from the VLSI dataset[4] within the World TSP collection, which include TSP instances with sizes ranging from 104,815 to 744,710 nodes.

### B.3 BASELINE METHODS

To evaluate generalization performance of a pre-trained model across both small- and large-scale TSPs, we select seven baseline models that have recently demonstrated SOTA performance across various scales. These models include 1) RL-based models: Omni-TSP (Zhou et al., 2023), ELG (Gao et al., 2024), INViT-3V (Fang et al., 2024), and UDC (Zheng et al., 2024); 2) SL-based models:

---

[3]URL: https://www.math.uwaterloo.ca/tsp/world/countries.html
[4]URL: https://www.math.uwaterloo.ca/tsp/vlsi/page11.html

LEHD (Luo et al., 2023) and BQ (Drakulic et al., 2023); and 3) SIL-based model: GD Pirnay & Grimm (2024). All baseline models were trained on a uniform distribution pattern, except for Omni-TSP, which was trained on diverse distribution patterns. Among these, UDC utilizes a D&C strategy, whereas the others are non-D&C neural TSP solvers. For the comparative experiments, we used the publicly available pre-trained parameters and default settings for all models, with two exceptions: For INViT, we adjust the configuration to handle multiple instances simultaneously, rather than the originally designed single-instance setup, to reduce execution time and ensure a fair comparison; For UDC, we set the hyperparameter values to $x$=250 and $\alpha$=1 in all relevant experiments. Furthermore, for a fair comparison in terms of computational efficiency, we report results only for the two baseline models (LEHD and BQ) combined with the greedy search strategy.

## B.4 Evaluation Matrices

For all baselines and GELD, we report the average gap to the (near-)optimal solutions. The solutions for synthetic datasets are computed using LKH3 (Helsgaun, 2017), while for real-world datasets, we use the best known solutions. To control a reasonable computing time consumption, TSP-{100, 500, 1000} instances are solved by LKH3 with 20000 iterations over 10 runs, whereas TSP-{5000, 10000} instances are solved by LKH3 with 20000 iterations over a single run. We present the solution length computed by LKH on the synthetic dataset in Table 6. The gap for each TSP instance is computed as follows:

$$\text{gap} = \frac{\text{L}(\pi^{model}) - \text{L}(\pi^{opt})}{\text{L}(\pi^{opt})} \times 100\%, \tag{15}$$

where $\pi^{model}$ denotes the solution produced by the model and $\pi^{opt}$ denotes the (near-)optimal solution. Furthermore, we report the inference time for each baseline method across all dataset. To ensure a fair comparison of inference time for the synthetic dataset, we intend to maintain an equal batch size for all models. However, due to the GPU memory constraint (24GB), we use the maximum batch size $n_{bs}$ that each model can solve simultaneously. This batch size, reflecting the model's parallel processing capability, serves as a practical measure of inference efficiency under real-world, resource-constrained conditions.

Table 6: Solution length computed by LKH3 on the synthetic dataset

|           | TSP-100 | TSP-500 | TSP-1000 | TSP-5000 | TSP-10000 |
|-----------|---------|---------|----------|----------|-----------|
| uniform   | 7.8693  | 16.5601 | 23.2215  | 50.9830  | 73.1436   |
| clustered | 5.3876  | 10.3447 | 14.0982  | 28.8359  | 40.2628   |
| explosion | 6.5397  | 12.0101 | 16.0543  | 31.9792  | 41.2801   |
| implosion | 7.1135  | 14.4128 | 20.1932  | 45.0435  | 63.7273   |

## B.5 Detailed Results on Real-world Datasets

We conduct a comprehensive evaluation of the baseline models and GELD on both TSPLIB and World TSP instances, as detailed in Tables 7 and 8, respectively. Additionally, the performance of baseline models, when integrated with GELD on the World TSP dataset, is presented in Table 9.

The largest TSP instance each baseline model can solve is as follows: Omni-TSP (10,639), LEHD (14,051), BQ (11,849), ELG (10,639), INViT-3V (33,708), GD (18,512), and UDC (10,639). Additionally, UDC failed to solve instances with fewer than 100 nodes due to unknown errors.

The results on real-world datasets and synthetic datasets demonstrate GELD outperforms all baseline models, including the SOTA D&C-based model UDC (Zheng et al., 2024). This superior performance can be attributed to GELD's effective integration of global and local information, whereas UDC is suboptimal in these experiments because it may overlook correlations between sub-problems.

Table 7: Detailed results (gap (%)) for all included TSPLIB instances

| Instance | UDC G* | GD G | INViT-3V G† | BQ G | LEHD G | ELG G* | Omni-TSP G* | GELD (Ours) G | GELD (Ours) BOTH |
|---|---|---|---|---|---|---|---|---|---|
| eil51 | - | 6.66 | 0.94 | 2.71 | 1.64 | 1.41 | 2.82 | 1.39 | **0.70** |
| berlin52 | - | 0.99 | 0.11 | 17.08 | 0.03 | **0.01** | 12.97 | 0.04 | 0.03 |
| st70 | - | 0.33 | 1.19 | 2.06 | 0.33 | **0.15** | 2.22 | 1.63 | 0.31 |
| pr76 | - | 0.99 | 0.36 | 0.11 | 0.22 | 0.69 | 2.45 | 0.13 | **0.00** |
| eil76 | - | 2.81 | 2.79 | 4.92 | 2.54 | 1.49 | 5.20 | 2.65 | **1.37** |
| rat99 | - | 0.91 | 1.57 | 18.49 | 1.10 | 4.54 | 13.13 | 0.96 | **0.68** |
| kroA100 | **0.02** | 0.13 | 0.42 | 12.15 | 0.12 | 1.67 | 9.07 | 0.43 | **0.02** |
| kroE100 | 0.50 | 0.07 | 1.15 | 13.63 | 0.43 | 2.21 | 5.12 | 0.57 | **0.00** |
| kroB100 | 0.18 | 0.45 | 0.26 | 4.35 | 0.26 | 1.65 | 12.78 | 0.31 | **0.00** |
| rd100 | 0.37 | 0.15 | 2.48 | 9.50 | 0.01 | 0.44 | 1.29 | 1.10 | **0.01** |
| kroD100 | 0.07 | 7.24 | 2.18 | 11.13 | 0.38 | 2.62 | 5.35 | 1.43 | **0.00** |
| kroC100 | **0.01** | 0.64 | 0.34 | 7.50 | 0.32 | 1.87 | 10.07 | **0.01** | 0.01 |
| eil101 | 2.81 | 3.57 | 3.82 | 4.77 | 2.31 | **0.64** | 3.82 | 2.38 | 2.07 |
| lin105 | **0.03** | 0.19 | 1.72 | 12.35 | 0.34 | 2.57 | 11.01 | 0.19 | **0.03** |
| pr107 | 0.65 | 4.89 | 1.22 | 13.74 | 11.24 | 3.60 | 3.66 | 4.39 | **0.00** |
| pr124 | 0.88 | 1.78 | 0.53 | 16.84 | 1.11 | 0.26 | 1.46 | 21.03 | **0.08** |
| bier127 | 1.09 | 2.04 | 2.79 | 6.30 | 4.76 | 4.70 | 8.34 | 7.55 | **0.01** |
| ch130 | **0.15** | 1.11 | 1.90 | 0.20 | 0.55 | 0.43 | 4.19 | 1.30 | 0.58 |
| pr136 | 0.42 | **0.24** | 1.97 | 9.87 | 0.45 | 2.28 | 1.04 | 2.42 | 1.74 |
| pr144 | 0.50 | 0.38 | 1.30 | 14.73 | **0.19** | 0.55 | 4.21 | 2.42 | 0.38 |
| kroA150 | **0.00** | 0.93 | 1.08 | 4.95 | 1.40 | 2.04 | 4.91 | 1.03 | 0.37 |
| kroB150 | 0.08 | 0.51 | 2.74 | 7.19 | 0.76 | 1.47 | 6.02 | **0.04** | **0.04** |
| ch150 | 0.37 | 0.70 | 2.10 | 5.64 | 0.52 | 1.10 | 2.45 | 0.89 | **0.04** |
| pr152 | 1.57 | 11.53 | 6.63 | 11.92 | 12.14 | **0.41** | 1.20 | 9.34 | 6.48 |
| u159 | 0.88 | 0.92 | 1.84 | **0.00** | 1.13 | 1.39 | 2.06 | 0.88 | 0.74 |
| rat195 | 0.92 | 2.25 | 2.80 | 10.93 | 1.42 | 6.11 | 19.80 | 1.50 | **0.82** |
| d198 | **4.44** | 10.34 | 10.44 | 10.31 | 9.23 | 14.23 | 14.25 | 13.25 | 6.46 |
| kroA200 | **0.06** | 1.13 | 1.49 | 8.79 | 0.64 | 2.09 | 6.46 | 0.84 | 0.16 |
| kroB200 | 0.20 | 0.39 | 2.86 | 10.74 | **0.16** | 1.58 | 9.25 | **0.16** | **0.16** |
| tsp225 | **0.00** | 0.46 | 1.53 | 4.70 | **0.00** | 4.52 | 8.48 | 0.16 | 0.00 |
| ts225 | 0.19 | 0.33 | 4.68 | 13.48 | 0.28 | 2.52 | 2.56 | 1.10 | **0.00** |
| pr226 | 0.30 | 0.62 | 3.73 | 11.75 | 1.11 | 1.43 | 2.01 | 10.72 | **0.01** |
| gil262 | 3.38 | **0.85** | 2.99 | 4.76 | 1.60 | 2.06 | 43.99 | 5.92 | 1.05 |
| pr264 | **0.15** | 16.89 | 3.47 | 12.50 | 5.48 | 5.66 | 6.17 | 17.40 | 9.48 |
| a280 | 2.95 | 2.34 | 3.88 | **0.46** | 3.02 | 5.93 | 8.72 | 2.03 | 1.02 |
| pr299 | 2.34 | 1.59 | 4.31 | 6.65 | 2.81 | 4.92 | 10.65 | 0.69 | **0.21** |
| lin318 | 7.10 | 1.98 | 3.16 | 10.36 | 1.41 | 4.42 | 8.17 | 1.53 | **0.97** |
| rd400 | 1.79 | 2.36 | 3.91 | 3.05 | 1.00 | 6.26 | 5.14 | 3.10 | **0.52** |
| f1417 | 7.24 | 33.66 | **4.99** | 19.01 | 7.76 | 7.55 | 15.15 | 20.75 | 7.77 |
| pr439 | 12.87 | 3.03 | 7.02 | 7.14 | 3.37 | 7.45 | 12.06 | 7.93 | **1.55** |
| pcb442 | 4.88 | 9.26 | 2.96 | 0.90 | 3.11 | 7.05 | 8.59 | 0.35 | **0.33** |
| d493 | 7.95 | 12.19 | 7.68 | 8.00 | 9.49 | 31.18 | 27.95 | 6.20 | **3.91** |
| u574 | 4.15 | 3.02 | 5.22 | 1.76 | 2.73 | 10.40 | 18.73 | 1.37 | **0.40** |
| rat575 | 7.78 | 8.98 | 4.36 | 10.07 | 3.02 | 9.49 | 21.48 | 2.24 | **0.77** |
| p654 | 33.07 | 22.37 | 10.78 | 16.03 | **3.30** | 4.32 | 14.60 | 10.04 | 6.41 |
| d657 | 10.25 | 4.81 | 8.91 | 8.62 | 8.05 | 11.36 | 15.09 | 9.02 | **1.77** |
| u724 | 3.75 | 4.88 | 3.86 | 2.18 | 3.27 | 10.35 | 19.35 | 1.96 | **0.86** |
| rat783 | 4.49 | 7.11 | 4.85 | 9.81 | 3.91 | 9.56 | 28.26 | 1.95 | **1.28** |
| pr1002 | **1.84** | 7.84 | 7.53 | 8.75 | 4.44 | 11.54 | 20.55 | 5.85 | 2.80 |
| u1060 | 9.23 | 18.00 | 6.39 | 8.63 | 10.00 | 12.18 | 31.32 | 12.33 | **2.87** |
| vm1084 | 3.75 | 22.47 | 6.24 | 10.39 | 5.42 | 15.81 | 25.62 | 3.47 | **1.18** |
| pcb1173 | 9.15 | 11.62 | 5.51 | 11.70 | 8.01 | 13.95 | 27.28 | 2.38 | **1.34** |
| d1291 | 12.90 | 22.51 | 13.16 | 11.13 | 14.13 | 9.39 | 32.43 | 12.44 | **4.62** |
| r11304 | 13.59 | 15.40 | 6.83 | 8.77 | 8.14 | 13.30 | 25.62 | 4.37 | **1.41** |

| | UDC | GD | INViT-3V | BQ | LEHD | ELG | Omni-TSP | GELD (Ours) | |
|---|---|---|---|---|---|---|---|---|---|
| Instance | $G^*$ | G | $G^\dagger$ | G | G | $G^*$ | $G^*$ | G | BOTH |
| r11323 | 9.73 | 18.19 | 6.75 | 7.64 | 9.26 | 12.42 | 29.76 | 12.59 | **2.27** |
| nrw1379 | 9.57 | 104.77 | 4.38 | 9.83 | 15.49 | 12.57 | 23.00 | 2.27 | **1.00** |
| f1400 | 25.11 | 84.65 | 11.89 | 31.19 | 18.80 | 8.74 | 18.18 | 23.12 | **7.15** |
| u1432 | 6.61 | 10.30 | 4.25 | 4.98 | 7.96 | 10.65 | 22.30 | 5.07 | **2.80** |
| f1577 | 23.75 | 65.74 | 7.53 | 21.61 | 14.68 | 8.35 | 32.75 | 9.44 | **5.15** |
| d1655 | 9.11 | 47.28 | 10.58 | 17.01 | 13.89 | 15.66 | 34.92 | 14.10 | **6.45** |
| vm1748 | 7.68 | 19.12 | 8.41 | 11.18 | 10.10 | 17.13 | 30.84 | 4.35 | **0.86** |
| u1817 | 8.39 | 28.70 | 6.90 | 9.43 | 10.32 | 12.62 | 39.72 | 9.43 | **3.08** |
| rl1889 | 22.28 | 26.59 | 9.08 | 14.91 | 7.49 | 17.12 | 37.50 | 6.32 | **3.41** |
| d2103 | 17.96 | 57.66 | 10.48 | 17.47 | 14.57 | 6.90 | 36.05 | 10.88 | **4.42** |
| u2152 | 13.55 | 32.67 | 7.20 | 9.08 | 12.65 | 12.12 | 43.01 | 8.68 | **5.16** |
| u2319 | 6.06 | 19.98 | 0.62 | 3.41 | 4.18 | 3.88 | 17.61 | 0.43 | **0.34** |
| pr2392 | 11.17 | 32.68 | 6.80 | 9.26 | 12.33 | 16.95 | 40.08 | 6.12 | **3.04** |
| pcb3038 | 7.14 | 35.92 | 7.05 | 13.44 | 13.44 | 16.75 | 40.08 | 8.63 | **2.73** |
| fl3795 | 40.23 | 331.22 | 11.29 | 32.09 | 13.55 | 13.46 | 54.24 | 21.26 | **10.66** |
| fn14461 | 17.29 | 134.34 | 5.58 | 21.38 | 19.05 | 15.98 | 47.99 | 12.38 | **2.99** |
| r15915 | 21.10 | 288.03 | 8.68 | 24.58 | 24.17 | 16.17 | 62.61 | 11.83 | **7.02** |
| r15934 | 31.41 | 363.20 | 10.00 | 30.17 | 24.11 | 18.08 | 63.94 | 11.68 | **7.17** |
| r111849 | 23.37 | 598.01 | 9.05 | 45.21 | 38.04 | OOM | OOM | 14.94 | **6.11** |
| usa13509 | OOM | 2252.54 | **8.23** | OOM | 71.11 | OOM | OOM | 17.39 | 8.97 |
| brd14051 | OOM | 700.75 | 7.40 | OOM | 41.22 | OOM | OOM | 17.32 | **4.17** |
| d15112 | OOM | 660.57 | 6.21 | OOM | OOM | OOM | OOM | 14.57 | **3.58** |
| d18512 | OOM | 744.35 | 6.99 | OOM | OOM | OOM | OOM | 15.26 | **6.64** |
| Avg. gap | 7.34 | 90.34 | 4.86 | 10.92 | 7.56 | 7.25 | 18.07 | 6.28 | **2.35** |
| Avg. time | 5.6s | 2.7m | 26.2s | 1.4m | 47.0s | 6.1s | 3.8s | 3.8s | 27.6s |

**End of Table**

Table 8: Detailed results (gap(%)) for all included National TSPs

| | UDC | GD | INViT-3V | BQ | LEHD | ELG | Omni-TSP | GELD (Ours) | |
|---|---|---|---|---|---|---|---|---|---|
| Instance | $G^*$ | G | $G^\dagger$ | G | G | $G^*$ | $G^*$ | G | BOTH |
| WI29 | - | 0.60 | 0.00 | 19.95 | 0.06 | 4.54 | 0.05 | 0.71 | **0.00** |
| DJ38 | - | 6.41 | 0.06 | 28.63 | 0.17 | **0.02** | 5.21 | 0.11 | 0.06 |
| QA194 | 0.58 | 236.40 | 2.88 | 12.18 | 27.15 | 7.06 | 10.44 | 0.53 | **0.02** |
| UY734 | 5.54 | 284.43 | 5.38 | 9.26 | 20.98 | 10.77 | 14.83 | 3.35 | **2.00** |
| ZI929 | 12.80 | 111.47 | 6.54 | 13.67 | 18.34 | 14.61 | 21.24 | 9.05 | **2.97** |
| LU980 | 11.78 | 2368.93 | 4.96 | 7.83 | 93.28 | 12.27 | 17.58 | 2.89 | **1.40** |
| RW1621 | 17.34 | 2722.21 | 7.42 | 12.79 | 58.53 | 11.42 | 27.20 | 7.99 | **3.61** |
| MU1979 | 15.41 | 1351.43 | 13.06 | 48.92 | 42.65 | 22.54 | 52.06 | 17.55 | **9.06** |
| NU3496 | 17.57 | 3616.20 | 10.74 | 22.12 | 84.94 | 17.20 | 44.75 | 10.91 | **3.58** |
| CA4663 | 22.39 | 684.98 | **9.47** | 78.38 | 40.22 | 88.57 | 162.60 | 22.77 | 11.64 |
| TZ6117 | 23.26 | 2007.00 | 9.45 | 32.11 | 51.24 | 20.69 | 59.20 | 14.68 | **5.64** |
| EG7146 | 27.11 | 1281.81 | 12.88 | 170.87 | 42.15 | 209.58 | 151.05 | 19.06 | **5.55** |
| YM7663 | 28.03 | 3632.62 | 13.37 | 82.37 | 93.84 | 60.12 | 79.25 | 18.06 | **7.17** |
| PM8079 | 20.85 | 7728.54 | 10.47 | 103.36 | 207.10 | 22.85 | 72.56 | 17.00 | **7.52** |
| EI8246 | 15.70 | 4568.96 | **7.09** | 39.19 | 131.26 | 20.73 | 61.70 | 14.98 | 7.73 |
| AR9152 | 29.17 | 1753.86 | 12.63 | 64.43 | 56.54 | 21.66 | 72.70 | 16.74 | **9.10** |
| JA9847 | 46.86 | 10429.63 | **12.20** | 197.73 | 132.74 | 37.71 | 80.34 | 23.58 | 12.61 |
| GR9882 | 19.44 | 2245.37 | 13.25 | 78.22 | 74.69 | 23.95 | 70.10 | 18.57 | **6.10** |
| KZ9976 | 18.58 | 1172.08 | 9.15 | 86.37 | 55.72 | 23.30 | 102.23 | 19.25 | **7.55** |
| FI10639 | 18.14 | 3709.73 | 10.04 | 55.65 | 98.52 | 22.44 | 71.67 | 14.62 | **6.53** |
| MO14185 | OOM | 3629.80 | 8.41 | OOM | OOM | OOM | OOM | 16.07 | **7.07** |
| HO14473 | OOM | 9842.77 | 13.13 | OOM | OOM | OOM | OOM | 18.35 | **8.16** |
| IT16862 | OOM | 3762.62 | 9.13 | OOM | OOM | OOM | OOM | 18.88 | **7.39** |

| Instance | UDC G* | GD G | INViT-3V G† | BQ G | LEHD G | ELG G* | Omni-TSP G* | GELD (Ours) G | BOTH |
|---|---|---|---|---|---|---|---|---|---|
| | | | | Continued from previous page | | | | | |
| VM22775 | OOM | OOM | 9.75 | OOM | OOM | OOM | OOM | 20.41 | **8.32** |
| SW24978 | OOM | OOM | 8.58 | OOM | OOM | OOM | OOM | 17.87 | **6.88** |
| BM33708 | OOM | OOM | **7.35** | OOM | OOM | OOM | OOM | 18.76 | 7.42 |
| CH71009 | OOM | OOM | OOM | OOM | OOM | OOM | OOM | 25.41 | **13.92** |
| Avg. gap | 19.44 | 2919.47 | 8.75 | 58.20 | 66.51 | 32.60 | 58.83 | 14.39 | **6.26** |
| Avg. time | 6.3s | 8.9m | 3.4m | 14.8m | 2.8m | 3.7m | 2.3m | 23.4s | 1.4m |

**End of Table**

Table 9: Detailed results (gap(%)) for all included National TSPs using
GELD (with PRC(1000)) as a post-processing method

| Instance | UDC +Ours | GD +Ours | INViT-3V +Ours | BQ +Ours | LEHD +Ours | ELG +Ours | Omni-TSP +Ours | Random Insertion - | Random Insertion +Ours |
|---|---|---|---|---|---|---|---|---|---|
| WI29 | - | 0.53 | 0.00 | 0.00 | 0.00 | 4.25 | 0.00 | 0.00 | 0.00 |
| DJ38 | - | 0.05 | 0.05 | 0.05 | 0.05 | 0.05 | 0.05 | 17.55 | 0.05 |
| QA194 | 0.58 | 4.52 | 0.68 | 4.52 | 7.12 | 0.67 | 0.67 | 11.54 | 2.35 |
| UY734 | 4.07 | 20.72 | 1.37 | 2.88 | 2.90 | 1.91 | 1.51 | 13.23 | 1.26 |
| ZI929 | 11.87 | 13.43 | 2.95 | 5.91 | 3.61 | 4.30 | 1.92 | 9.35 | 2.68 |
| LU980 | 6.83 | 33.83 | 2.42 | 1.64 | 3.21 | 1.48 | 1.39 | 12.00 | 1.30 |
| RW1621 | 12.71 | 70.19 | 1.98 | 1.26 | 4.84 | 1.74 | 3.55 | 12.48 | 1.41 |
| MU1979 | 11.33 | 70.43 | 7.22 | 14.58 | 8.65 | 7.27 | 5.34 | 9.09 | 2.39 |
| NU3496 | 7.82 | 94.52 | 3.65 | 4.77 | 10.14 | 4.39 | 4.53 | 13.58 | 3.83 |
| CA4663 | 13.25 | 27.96 | 5.43 | 7.42 | 9.97 | 21.71 | 8.54 | 14.81 | 4.94 |
| TZ6117 | 13.44 | 87.03 | 3.89 | 8.73 | 8.15 | 6.12 | 4.64 | 14.42 | 2.76 |
| EG7146 | 20.14 | 60.27 | 6.56 | 7.93 | 7.58 | 63.24 | 18.72 | 14.35 | 4.07 |
| YM7663 | 17.43 | 207.35 | 8.59 | 10.19 | 10.82 | 25.99 | 7.24 | 13.79 | 3.68 |
| PM8079 | 9.79 | 203.51 | 2.53 | 7.15 | 11.90 | 8.41 | 5.83 | 12.07 | 3.12 |
| EI8246 | 9.95 | 188.51 | 3.02 | 7.71 | 10.70 | 7.92 | 4.24 | 14.14 | 3.31 |
| AR9152 | 18.64 | 141.87 | 7.46 | 9.18 | 8.39 | 9.16 | 6.23 | 13.73 | 3.58 |
| JA9847 | 34.18 | 135.25 | 5.58 | 15.65 | 7.08 | 16.33 | 5.94 | 12.80 | 3.89 |
| GR9882 | 13.62 | 97.99 | 7.13 | 9.78 | 9.43 | 8.22 | 3.03 | 12.02 | 1.92 |
| KZ9976 | 10.89 | 58.46 | 3.36 | 9.94 | 9.68 | 7.62 | 8.30 | 14.02 | 3.32 |
| FI10639 | 11.19 | 158.62 | 5.49 | 8.81 | 11.64 | 7.99 | 5.40 | 13.63 | 3.00 |
| MO14185 | - | 106.41 | 3.41 | - | - | - | - | 13.45 | 2.86 |
| HO14473 | - | 194.83 | 7.69 | - | - | - | - | 11.68 | 3.11 |
| IT16862 | - | 103.73 | 4.71 | - | - | - | - | 13.77 | 2.66 |
| VM22775 | - | - | 5.50 | - | - | - | - | 12.44 | 2.19 |
| SW24978 | - | - | 3.55 | - | - | - | - | 13.87 | 3.02 |
| BM33708 | - | - | 2.81 | - | - | - | - | 13.72 | 2.77 |
| CH71009 | - | - | - | - | - | - | - | 14.21 | 3.61 |
| Avg. gap | 12.66 | 90.43 | 4.12 | 6.91 | 7.29 | 10.44 | 4.85 | 12.66 | 2.71 |
| Gain(%) | **35.05** | **96.90** | **52.91** | **88.13** | **89.04** | **67.98** | **91.76** | **78.59** | |
| Avg. time | + 26.2s | +28.3s | +34.0s | +26.6s | +26.6s | +26.6s | +26.6s | 1.1s | +36.8s |

**End of Table**

## B.6 COMPARATIVE ANALYSIS WITH HEATMAP-BASED MODELS AND HEURISTIC ALGORITHMS

In this subsection, we conduct additional comparative experiments involving heatmap-based models and heuristic algorithms. Specifically, we select DIFUSCO (Sun & Yang, 2023) and the nearest neighbor+2-opt method as representatives of heatmap-based models and heuristic algorithms, respectively. For DIFUSCO, we utilize its publicly available pre-trained parameters (trained on 100-node instances) and adopt its default inference settings: a sampling number of 4 and 5000 iterations

for the 2-opt optimization. For the nearest neighbor+2-opt algorithm, the solution generated by the nearest neighbor algorithm serves as the initial solution, followed by 2-opt optimization with 1000 iterations.

We present the comparison results on the synthetic TSP instances of the uniform distribution in Table 10. As shown, the performance of DIFUSCO is heavily dependent on the iterative optimization process of 2-opt that (often) specifically tailored to TSP, while our method does not. More importantly, our proposed GELD + BOTH outperforms DIFUSCO (Sun & Yang, 2023) and the nearest neighbor+2-opt heuristic across all problem sizes in terms of both solution quality and computational efficiency.

Table 10: Performance comparisons with DIFUSCO and the nearest neighbor + 2-opt algorithm on synthetic TSP instances of the uniform distribution

| Method | TSP-100 (200) | | TSP-500 (200) | | TSP-1000 (200) | | TSP-5000 (20) | | TSP-10000 (20) | | Average |
| | gap(%)↓ | time↓, $n_{bs}$ ↑ | gap(%)↓ | time↓, $n_{bs}$ ↑ | gap(%)↓ | time↓, $n_{bs}$ ↑ | gap(%)↓ | time↓, $n_{bs}$ ↑ | gap(%)↓ | time↓, $n_{bs}$ ↑ | gap(%)↓ |
|---|---|---|---|---|---|---|---|---|---|---|---|
| Nearest neighbor + 2-opt | 5.69 | 3.7s, 1 | 5.55 | 9.4s, 1 | 5.24 | 34.9s, 1 | 5.53 | 2.7m, 1 | 4.32 | 13.5m, 1 | 5.27 |
| DIFUSCO + S + 2-opt | 0.06 | 2.0m, 1 | 3.95 | 22.1m, 1 | 3.33 | 1.6h, 1 | 6.54 | 37.8m, 1 | 4.72 | 2.1h, 1 | 3.72 |
| GELD + G | 1.11 | 0.6s, 200 | 2.39 | 1.8s, 200 | 2.94 | 3.6s, 200 | 7.62 | 10.8s, 20 | 9.33 | 21.6s, 20 | 4.68 |
| GELD + BOTH | **0.06** | 19.2s, 200 | **0.52** | 1.6m, 200 | **0.58** | 3.7m, 200 | **2.77** | 1.8m, 20 | **2.38** | 3.9m, 20 | **1.26** |

Symbol "S" denotes the sampling operation used in Sun & Yang (2023).

### B.7 SENSITIVITY ANALYSIS

In this subsection, we conduct ablation studies to evaluate the sensitivity of two hyperparameters: the number of regions $m$ and the range of local selection $k_m$. To examine the impact of $m$, we use the first-stage training for GELD with three configurations, testing its effect on model performance across various TSP sizes and distributions: 1) $m_r$=2, $m_c$=2, resulting in $m = 4$ regions; 2) $m_r$=3, $m_c$=3, resulting in $m = 9$ regions; and 3) $m_r$=4, $m_c$=4, resulting in $m = 16$ regions.

The results presented in Table 11 indicate that increasing $m$ generally improves model performance albeit with a slight reduction in inference speed. Additionally, the variations in $m$ have minimal impact on overall performance, demonstrating the model's robustness across different configurations.

For the range of local selection $k$, we test three values (50, 100, 150) on the synthetic TSP instances of the uniform distribution. As shown in Table 12, larger values of $k$ improve model performance while decreasing inference speed. These findings further highlight the importance of adopting LD in our model design.

### B.8 PERFORMANCE OF GELD ON OTHER PUBLICLY AVAILABLE SYNTHETIC DATASETS

In this subsection, we evaluate the performance of GELD using publicly available synthetic datasets. Specifically, three datasets are employed: Dataset 1[5] (used by T2T (Li et al., 2023)), Dataset 2[6] (used by DIMES (Qiu et al., 2022)), and Dataset 3[7] (used by Att-GCN (Fu et al., 2021) and DIFUSCO (Sun & Yang, 2023)). The results of GELD's performance on these datasets are presented in Tables 13, 14, and 15, respectively. As shown, our proposed GELD achieves excellent performance across all datasets.

## C SOLUTION VISUALIZATIONS

In this section, we present a visualization of three TSP solutions in the World TSP dataset. Specifically, we select DJ38, TZ6117, and FI10639 as representatives of small-, medium- and large-scale TSP instances, respectively. The solutions for these instances are illustrated in Figures 4, 5 and

---

[5]URL: https://github.com/Thinklab-SJTU/T2TCO/tree/main/data/tsp

[6]URL: https://github.com/DIMESTeam/DIMES/tree/main/TSP/data

[7]URL: https://github.com/Spider-scnu/TSP/tree/master/MCTS

Table 11: Performance of the first-stage trained GELD with different $m$

| | Value | Inference | TSP-100 | | TSP-500 | | TSP-1000 | | TSP-5000 | | TSP-10000 | | Average |
|---|---|---|---|---|---|---|---|---|---|---|---|---|---|
| | | | gap(%)↓ | time↓ | gap(%)↓ | time↓ | gap(%)↓ | time↓ | gap(%)↓ | time↓ | gap(%)↓ | time↓ | gap(%)↓ |
| uniform | 4 | G | 0.71 | 0.6s | 3.81 | 1.7s | 4.81 | 3.4s | 15.75 | 10.3s | 17.08 | 20.8s | 8.43 |
| | | BOTH | 0.05 | 19.0s | 0.75 | 1.5m | 1.23 | 3.6m | 4.1 | 1.7m | 3.69 | 3.8m | 1.96 |
| | 9 | G | 0.86 | 0.6s | 3.28 | 1.8s | 4.17 | 3.6s | 13.61 | 10.8s | 15.21 | 21.6s | 7.43 |
| | | BOTH | 0.05 | 19.2s | 0.69 | 1.6m | 1.14 | 3.7m | 3.73 | 1.8m | 3.1 | 3.9m | 1.74 |
| | 16 | G | 0.94 | 0.6s | 3.44 | 1.9s | 5.15 | 3.9s | 12.68 | 11.4s | 12.15 | 23.4s | 6.87 |
| | | BOTH | 0.05 | 20.4s | 0.78 | 1.7m | 1.35 | 3.8m | 3.86 | 2.0m | 2.87 | 4.1m | 1.78 |
| clustered | 4 | G | 3.49 | 0.6s | 7.4 | 1.7s | 10.83 | 3.4s | 20.59 | 10.3s | 32.15 | 20.8s | 14.89 |
| | | BOTH | 0.51 | 19.0s | 1.86 | 1.5m | 3.20 | 3.6m | 5.81 | 1.7m | 5.48 | 3.8m | 3.37 |
| | 9 | G | 2.80 | 0.6s | 6.73 | 1.8s | 10.34 | 3.6s | 16.36 | 10.8s | 15.31 | 21.6s | 10.31 |
| | | BOTH | 0.49 | 19.2s | 1.87 | 1.6m | 3.24 | 3.7m | 5.42 | 1.8m | 4.01 | 3.9m | 3.01 |
| | 16 | G | 2.65 | 0.6s | 6.9 | 1.9s | 9.76 | 3.9s | 14.63 | 11.4s | 16.06 | 23.4s | 10.00 |
| | | BOTH | 0.42 | 20.4s | 1.66 | 1.7m | 2.79 | 3.8m | 5.36 | 2.0m | 4.23 | 4.1m | 2.89 |
| explosion | 4 | G | 1.28 | 0.6s | 5.52 | 1.7s | 9.37 | 3.4s | 18.79 | 10.3s | 28.56 | 20.8s | 12.70 |
| | | BOTH | 0.10 | 19.0s | 1.52 | 1.5m | 2.35 | 3.6m | 6.28 | 1.7m | 5.31 | 3.8m | 3.11 |
| | 9 | G | 1.00 | 0.6s | 4.68 | 1.8s | 8.02 | 3.6s | 17.11 | 10.8s | 17.87 | 21.6s | 9.74 |
| | | BOTH | 0.13 | 19.2s | 1.25 | 1.6m | 2.09 | 3.7m | 5.57 | 1.8m | 5.01 | 3.9m | 2.81 |
| | 16 | G | 1.34 | 0.6s | 5.9 | 1.9s | 9.59 | 3.9s | 14.71 | 11.4s | 17.51 | 23.4s | 9.81 |
| | | BOTH | 0.27 | 20.4s | 1.5 | 1.7m | 2.5 | 3.8m | 5.55 | 2.0m | 5.05 | 4.1m | 2.97 |
| implosion | 4 | G | 1.85 | 0.6s | 5.47 | 1.7s | 8.07 | 3.4s | 18.1 | 10.3s | 21.36 | 20.8s | 10.97 |
| | | BOTH | 0.17 | 19.0s | 1.37 | 1.5m | 2.17 | 3.6m | 5.18 | 1.7m | 4.3 | 3.8m | 2.64 |
| | 9 | G | 1.84 | 0.6s | 5.2 | 1.8s | 6.61 | 3.6s | 14.84 | 10.8s | 15.69 | 21.6s | 8.84 |
| | | BOTH | 0.17 | 19.2s | 1.35 | 1.6m | 1.9 | 3.7m | 4.39 | 1.8m | 3.78 | 3.9m | 2.32 |
| | 16 | G | 1.86 | 0.6s | 5.05 | 1.9s | 7.37 | 3.9s | 13.89 | 11.4s | 12.92 | 23.4s | 8.22 |
| | | BOTH | 0.23 | 20.4s | 1.44 | 1.7m | 2.23 | 3.8m | 4.62 | 2.0m | 3.27 | 4.1m | 2.36 |

Table 12: Performance of GELD with different $k_m$ on synthetic TSP instances of the uniform distribution

| Value | Inference | TSP-100 | | TSP-500 | | TSP-1000 | | TSP-5000 | | TSP-10000 | | Average |
|---|---|---|---|---|---|---|---|---|---|---|---|---|
| | | gap(%)↓ | time↓ | gap(%)↓ | time↓ | gap(%)↓ | time↓ | gap(%)↓ | time↓ | gap(%)↓ | time↓ | gap(%)↓ |
| 50 | G | 2.00 | 0.4s | 4.02 | 1.2s | 5.00 | 2.9s | 10.19 | 9.2s | 11.03 | 18.9s | 6.45 |
| | BOTH | 0.17 | 18.7s | 0.85 | 1.4m | 1.06 | 2.9m | 3.20 | 1.5m | 3.15 | 3.5m | 1.69 |
| 100 | G | 1.11 | 0.6s | 2.39 | 1.8s | 2.94 | 3.6s | 7.62 | 10.8s | 9.33 | 21.6s | 4.68 |
| | BOTH | 0.06 | 19.2s | 0.52 | 1.6m | 0.58 | 3.7m | 2.77 | 1.8m | 2.38 | 3.9m | 1.26 |
| 150 | G | 1.11 | 0.7s | 2.28 | 2.4s | 2.33 | 5.4s | 7.05 | 12.1s | 9.52 | 24.1s | 4.46 |
| | BOTH | 0.06 | 21.6s | 0.45 | 2.0m | 0.49 | 4.3m | 3.11 | 2.4m | 2.04 | 4.9m | 1.23 |

6, respectively. In each Figure, panel (a) shows the optimal solution (i.e., the best-known solution), and panels (b), (c), (d), (e), (f) show the solution produced by LEHD (G), INViT-3V (G†), GELD (BOTH), Random insertion, and Random insertion + GELD (PRC(1000)), respectively.

# D    LICENSES FOR USED RESOURCES

Table 16 summarizes the open-source codes and datasets used in this study, all of which are freely available for academic purposes.

Table 13: Performance of GELD on publicly available dataset 1

| Method | TSP-50 (1280) | | | TSP-100 (1280) | | | TSP-500 (128) | | | TSP-1000 (128) | | |
|---|---|---|---|---|---|---|---|---|---|---|---|---|
| | Length↓ | gap(%)↓ | time↓, $n_{bs}$ ↑ | Length↓ | gap(%)↓ | time↓, $n_{bs}$ ↑ | Length↓ | gap(%)↓ | time↓, $n_{bs}$ ↑ | Length↓ | gap(%)↓ | time↓, $n_{bs}$ ↑ |
| Concorde | 5.6876 | - | - | 7.7559 | - | - | 16.5458 | - | - | 23.1181 | - | - |
| GELD + G | 5.7467 | 1.04 | 0.6s, 1280 | 7.8377 | 1.05 | 1.2s, 1280 | 16.9601 | 2.50 | 1.2s, 128 | 23.9285 | 3.50 | 2.4s, 128 |
| GELD + BOTH | 5.6889 | 0.02 | 35.4s, 1280 | 7.7604 | 0.06 | 1.6m, 1280 | 16.6334 | 0.53 | 1.1m, 128 | 23.3209 | 0.88 | 2.2m, 128 |

Table 14: Performance of GELD on publicly available dataset 2

| Method | TSP-100 (10000) | | | TSP-500 (128) | | | TSP-1000 (128) | | | TSP-10000 (16) | | |
|---|---|---|---|---|---|---|---|---|---|---|---|---|
| | Length↓ | gap(%)↓ | time↓, $n_{bs}$ ↑ | Length↓ | gap(%)↓ | time↓, $n_{bs}$ ↑ | Length↓ | gap(%)↓ | time↓, $n_{bs}$ ↑ | Length↓ | gap(%)↓ | time↓, $n_{bs}$ ↑ |
| Concorde/LKH3 | 7.7645 | - | - | 16.5836 | - | - | 23.2268 | - | - | 71.7700 | - | - |
| GELD + G | 7.8419 | 1.00 | 9.1s, 1000 | 16.9601 | 2.27 | 1.2s, 128 | 23.9285 | 3.02 | 2.4s, 128 | 79.7566 | 11.13 | 18.2s, 16 |
| GELD + BOTH | 7.7659 | 0.02 | 11.9m, 1000 | 16.6334 | 0.30 | 1.1m, 128 | 23.3209 | 0.41 | 2.2m, 128 | 75.1468 | 4.71 | 3.2m, 16 |

Table 15: Performance of GELD on publicly available dataset 3

| Method | TSP-20 (10000) | | | TSP-50 (10000) | | | TSP-100 (10000) | | | TSP-200 (128) | | |
|---|---|---|---|---|---|---|---|---|---|---|---|---|
| | Length↓ | gap(%)↓ | time↓, $n_{bs}$ ↑ | Length↓ | gap(%)↓ | time↓, $n_{bs}$ ↑ | Length↓ | gap(%)↓ | time↓, $n_{bs}$ ↑ | Length↓ | gap(%)↓ | time↓, $n_{bs}$ ↑ |
| Concorde/LKH3 | 3.8306 | - | - | 5.6918 | - | - | 7.7645 | - | - | 10.7280 | - | - |
| GELD + G | 3.8838 | 1.39 | 0.6s, 10000 | 5.7502 | 1.03 | 3.0s, 2500 | 7.8419 | 1.00 | 10.8s, 1250 | 10.9159 | 1.75 | 0.6s, 128 |
| GELD + BOTH | 3.8306 | 0.00 | 1.8m, 10000 | 5.6919 | 0.00 | 5.0m, 2500 | 7.7659 | 0.02 | 12.3m, 1250 | 10.7485 | 0.19 | 0.4m, 128 |

| Method | TSP-500 (128) | | | TSP-1000 (128) | | | TSP-10000 (16) | | |
|---|---|---|---|---|---|---|---|---|---|
| | Length↓ | gap(%)↓ | time↓, $n_{bs}$ ↑ | Length↓ | gap(%)↓ | time↓, $n_{bs}$ ↑ | Length↓ | gap(%)↓ | time↓, $n_{bs}$ ↑ |
| Concorde/LKH3 | 16.5836 | - | - | 23.2268 | - | - | 71.7700 | - | - |
| GELD + G | 16.9601 | 2.27 | 1.2s, 128 | 23.9285 | 3.02 | 18.3s, 128 | 79.7566 | 11.13 | 18.6s, 16 |
| GELD + BOTH | 16.6334 | 0.30 | 1.1m, 128 | 23.3209 | 0.41 | 3.2m, 128 | 75.1468 | 4.71 | 3.2m, 16 |

Table 16: List of licenses for the codes and datasets used in this work

| Resource | Type | URL | License |
|---|---|---|---|
| LKH3 | Code | http://webhotel4.ruc.dk/ keld/research/LKH-3/ | Available for academic research use |
| Omni-TSP | Code | https://github.com/RoyalSkye/Omni-VRP | MIT License |
| BQ | Code | https://github.com/naver/bq-nco | CC BY-NC-SA 4.0 license |
| LEHD | Code | https://github.com/CIAM-Group/NCO_code/tree/main/single_objective/LEHD | Available for any non-commercial use |
| ELG | Code | https://github.com/gaocrr/ELG | MIT License |
| INViT-3V | Code | https://github.com/Kasumigaoka-Utaha/INViT | Available for academic research use |
| GD | Code | https://github.com/grimmlab/gumbeldore | Available for academic research use |
| UDC | Code | https://github.com/CIAM-Group/NCO_code/tree/main/single_objective/UDC-Large-scale-CO-master | Available for academic research use |
| TSPLib | Dataset | https://comopt.ifi.uni-heidelberg.de/software/TSPLIB95/ | Available for academic research use |
| World TSP | Dataset | https://www.math.uwaterloo.ca/tsp/world/index.html | Available for academic research use |

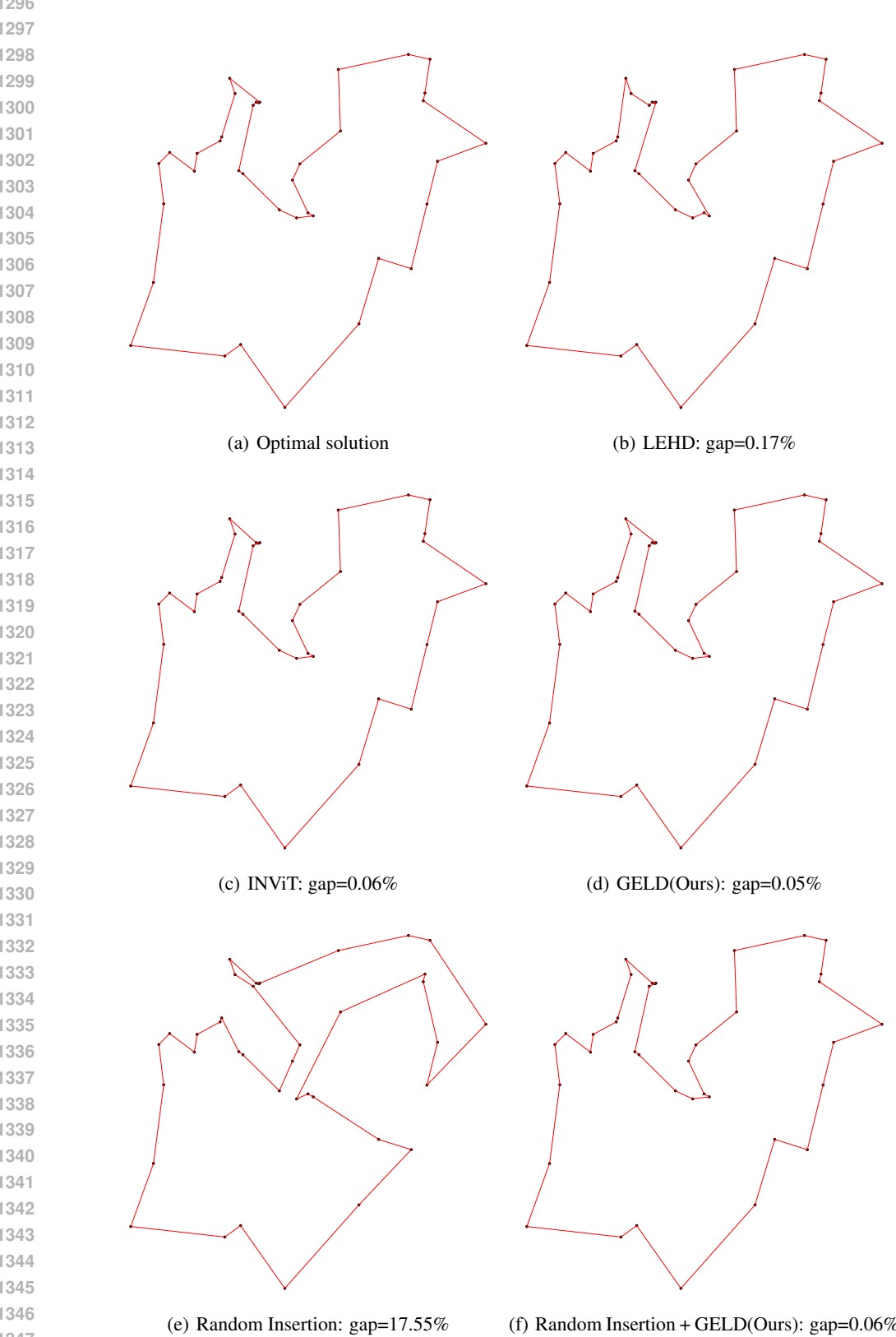

(a) Optimal solution

(b) LEHD: gap=0.17%

(c) INViT: gap=0.06%

(d) GELD(Ours): gap=0.05%

(e) Random Insertion: gap=17.55%

(f) Random Insertion + GELD(Ours): gap=0.06%

Figure 4: Visualization of solutions on DJ38 (small-scale) TSP instance with 38 nodes.

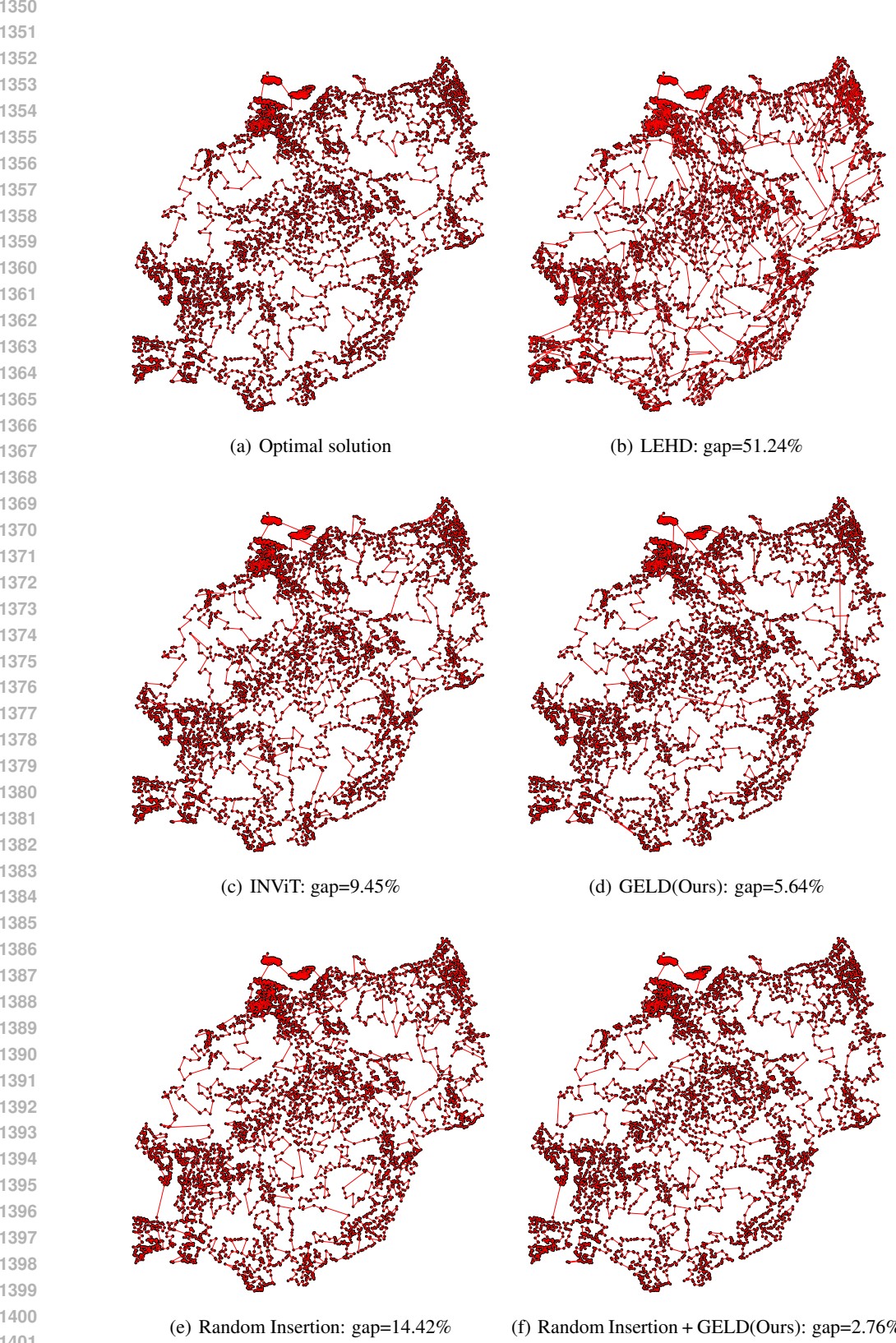

(a) Optimal solution

(b) LEHD: gap=51.24%

(c) INViT: gap=9.45%

(d) GELD(Ours): gap=5.64%

(e) Random Insertion: gap=14.42%

(f) Random Insertion + GELD(Ours): gap=2.76%

Figure 5: Visualization of solutions on TZ6117 (medium-scale) TSP instance with 6117 nodes.

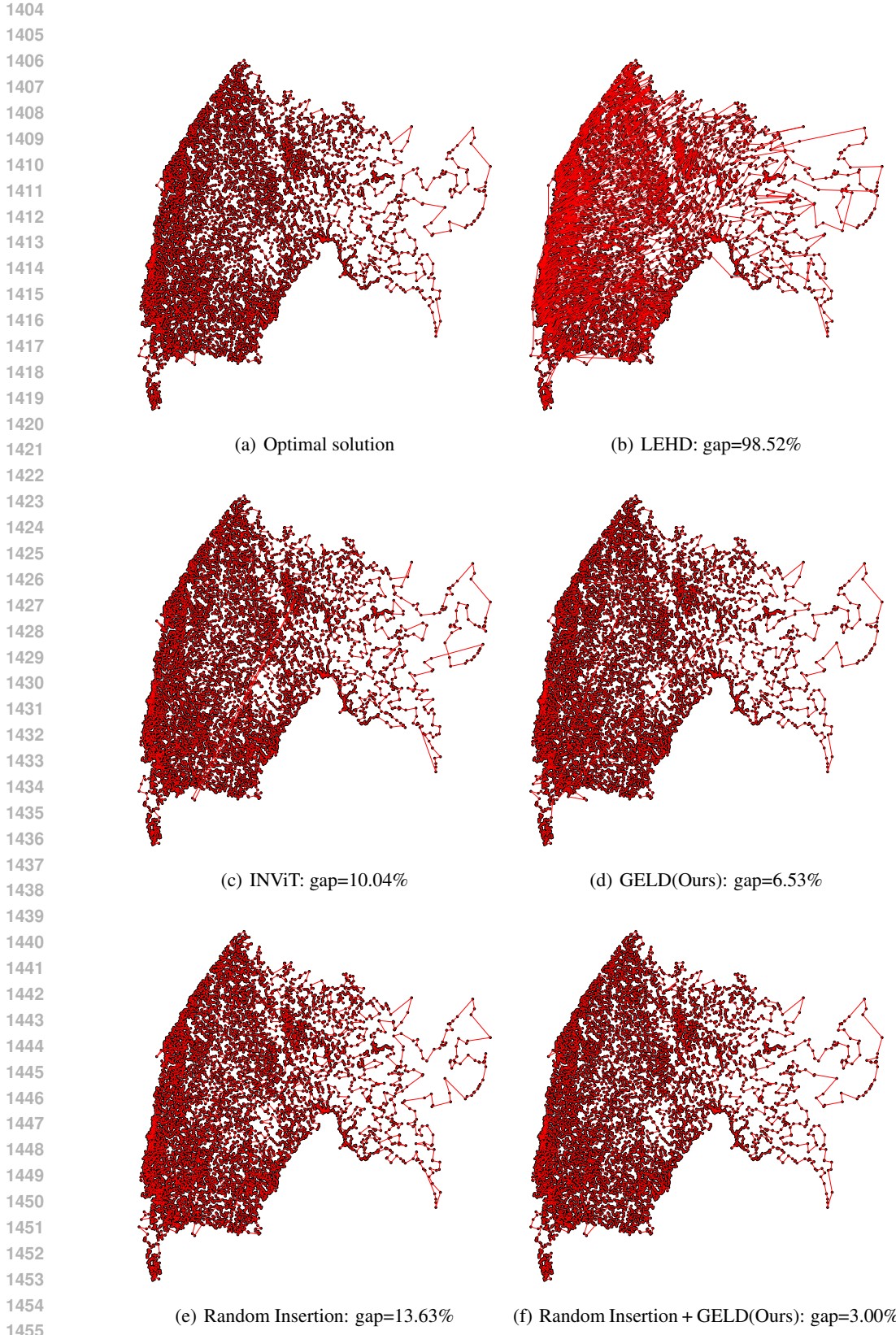

(a) Optimal solution

(b) LEHD: gap=98.52%

(c) INViT: gap=10.04%

(d) GELD(Ours): gap=6.53%

(e) Random Insertion: gap=13.63%

(f) Random Insertion + GELD(Ours): gap=3.00%

Figure 6: Visualization of solutions on FI10639 (large-scale) TSP instance with 10639 nodes.

