# OpenReview forum: "From Global Assessment to Local Selection: Efficiently Solving Traveling Salesman Problems of All Sizes"
_ICLR.cc/2025/Conference — Submitted to ICLR 2025_

### Official Review · Reviewer_6ZAK · 2024-10-30

**Soundness:** 2
**Presentation:** 2
**Contribution:** 2
**Rating:** 6
**Confidence:** 3

**Summary:**

Neural TSP solvers struggle to generalize across different TSP sizes and often have high computational complexity. The goal is to develop a unified model that can solve both small-scale and large-scale TSPs quickly and improve solution quality. It combines a Global-view Encoder (GE) and a Local-view Decoder (LD). GE uses a novel Region-Average Linear Attention (RALA) mechanism with low time-space complexity to capture global information. LD autoregressively selects the next node within a local selection range. A two-stage training approach is proposed. The first stage uses supervised learning on small-scale instances, and the second stage applies self-improvement learning on larger instances with a curriculum learning strategy. GELD outperforms seven state-of-the-art models in terms of solution quality and inference speed on both synthetic and real-world datasets. It can handle all test instances, while some baselines fail due to GPU memory constraints. It can also solve extremely large TSPs (up to 744,710 nodes) when integrated with heuristic algorithms.

**Strengths:**

The combination of a lightweight Global-view Encoder (GE) with a heavyweight Local-view Decoder (LD) and the use of Region-Average Linear Attention (RALA) in GE allow for efficient handling of global and local information, addressing the limitations of previous models that focused either too much on global or local aspects.

**Weaknesses:**

Although the hyperparameters are set based on certain considerations, a more in-depth exploration could be beneficial. For example, the impact of different values of m_r  and m_c  (the numbers of rows and columns for partitioning in RALA) on the model's performance for different TSP sizes and distributions could be further investigated.

The comparison with traditional heuristics is limited. While Random Insertion is used as a baseline in some experiments, other well-known heuristics (e.g., Lin-Kernighan, 2-opt) could be included in a more comprehensive comparison. This would provide a better understanding of how GELD compares to traditional methods in terms of solution quality and computational efficiency.

**Questions:**

1） In the two-stage training strategy, why was the maximum training size n_max set to 1000? What was the rationale behind this choice, and how did you explore other possible values?

2) The Region-Average Linear Attention (RALA) mechanism is a key innovation in the Global-view Encoder (GE). However, it is not clear how the choice of partitioning the nodes into regions (controlled by m_r and m_c) affects the balance between capturing global and local information. Can you provide more insights into this?

3) The Local-view Decoder (LD) restricts the candidate set to the k-nearest neighbors. How was the value of k determined, and what is the sensitivity of the model's performance to changes in k?

---

> ### Author Response · Authors · 2024-11-24
> **Response to Reviewer 6ZAK (Part 1)**
>
> Thank you for your review comments and providing us the insightful and critical feedback. Here are our detailed responses to your comments. We are more than happy to discuss any further concerns or questions that you may have.
>
> ---
>
> ### Weakness 1:
> >Although the hyperparameters are set based on certain considerations, a more in-depth exploration could be beneficial. For example, the impact of different values of m_r and m_c (the numbers of rows and columns for partitioning in RALA) on the model's performance for different TSP sizes and distributions could be further investigated.
>
> ### Response 3.1 (sensitivity analysis on hypeprarameter $m$):
>
> Thank you for your valuable suggestion. The hyperparameter $m$ divides a TSP topology into $m$ regions. The rationale behind this design is to strive for a good balance between model performance and computational complexity. As we mentioned in Section 4.1 and Appendix B.1, respectively:
> > "the hyperparameter $m=m_r\cdot m_c$ determines the granularity of the regional view: A larger *m*  allows the model to capture more detailed local features but increases computational complexity.
> > To balance performance and computational complexity, we set the numbers of rows and columns to $m_r$ = 3 and $m_c$ = 3, respectively, resulting in $m$ = 3 × 3 = 9 regions."
>
> Per your suggestion, we have now conducted a sensitivity analysis on $m$ by retraining GELD (first-stage training) with three configurations:
>
> - $m_r=2, m_c=2, m=4$ regions
> - $m_r=3, m_c=3, m=9$ regions
> - $m_r=4, m_c=4, m=16$ regions
>
> Notably, we set the numbers of rows and columns equal to reserve the TSP's square topology, thereby preventing potential biases from specific distribution patterns that occupy only a subset of the TSP topology (see Figure 3).
>
> The results on TSPs of the uniform and clustered distribution are as follows (see Table 10 of the revised manuscript for all results):
>
> |                |  |  | TSP-100          | TSP-500          | TSP-1000         | TSP-5000         | TSP-10000        | |
> |----------------|-------|-----------|------------------|------------------|------------------|------------------|------------------|----------------|
> | distrubution |   Value  |Inference |gap(%)↓, time↓   | gap(%)↓, time↓     | gap(%)↓, time↓   |gap (%)↓, time↓|gap(%)↓, time↓|      Average gap(%)↓   |
> | **uniform**    | 4     | G         | 0.71, 0.6s       | 3.81, 1.7s       | 4.81, 3.4s       | 15.75, 10.3s     | 17.08, 20.8s     | 8.43           |
> |                |       | BOTH      | 0.05, 19.0s      | 0.75, 1.5m       | 1.23, 3.6m       | 4.1, 1.7m        | 3.69, 3.8m       | 1.96           |
> |                | 9     | G         | 0.86, 0.6s       | 3.28, 1.8s       | 4.17, 3.6s       | 13.61, 10.8s     | 15.21, 21.6s     | 7.43           |
> |                |       | BOTH      | 0.05, 19.2s      | 0.69, 1.6m       | 1.14, 3.7m       | 3.73, 1.8m       | 3.1, 3.9m        | 1.74           |
> |                | 16    | G         | 0.94, 0.6s       | 3.44, 1.9s       | 5.15, 3.9s       | 12.68, 11.4s     | 12.15, 23.4s     | 6.87           |
> |                |       | BOTH      | 0.05, 20.4s      | 0.78, 1.7m       | 1.35, 3.8m       | 3.86, 2.0m       | 2.87, 4.1m       | 1.78           |
> | **clustered**  | 4     | G         | 3.49, 0.6s       | 7.4, 1.7s        | 10.83, 3.4s      | 20.59, 10.3s     | 32.15, 20.8s     | 14.89          |
> |                |       | BOTH      | 0.51, 19.0s      | 1.86, 1.5m       | 3.20, 3.6m       | 5.81, 1.7m       | 5.48, 3.8m       | 3.37           |
> |                | 9     | G         | 2.80, 0.6s       | 6.73, 1.8s       | 10.34, 3.6s      | 16.36, 10.8s     | 15.31, 21.6s     | 10.31          |
> |                |       | BOTH      | 0.49, 19.2s      | 1.87, 1.6m       | 3.24, 3.7m       | 5.42, 1.8m       | 4.01, 3.9m       | 3.01           |
> |                | 16    | G         | 2.65, 0.6s       | 6.9, 1.9s        | 9.76, 3.9s       | 14.63, 11.4s     | 16.06, 23.4s     | 10.00          |
> |                |       | BOTH      | 0.42, 20.4s      | 1.66, 1.7m       | 2.79, 3.8m       | 5.36, 2.0m       | 4.23, 4.1m       | 2.89           |
>
>
> The results indicate that increasing $m$ generally improves model performance with a slightly lengthened inference time. Hence, our model is shown as robust given the consistently high overall performance obtained with various settings of $m$.

---

> > ### Author Response · Authors · 2024-11-24
> > **Response to Reviewer 6ZAK (Part 2)**
> >
> > ### Weakness 2:
> > >The comparison with traditional heuristics is limited. While Random Insertion is used as a baseline in some experiments, other well-known heuristics (e.g., Lin-Kernighan, 2-opt) could be included in a more comprehensive comparison. This would provide a better understanding of how GELD compares to traditional methods in terms of solution quality and computational efficiency.
> >
> > ### Response 3.2 (limited comparison with traditional heuristics):
> > Thank you for your valuable feedback. Prior studies [1-2] have demonstrated that deep learning-based TSP solvers generally outperform traditional heuristics with the exception of LKH3, while offering significantly faster inference speed. Consequently, many recent studies [3-5] have also omitted comparisons with traditional heuristics. Following this convention, we did not include such comparisons in the previously submitted manuscript.
> >
> > It is worth noting that (near-)optimal solutions for the synthetic dataset in our experiments were generated using LKH3. As shown in Table 1, while LKH3 achieves superior solution quality compared to all models, it is considerably slower in terms of inference speed.
> >
> > Nonetheless, per your suggestion, we have now conducted experiments using the nearest neighbor+2-opt method (see Appendix B.6 of the revised manuscript for details). The results on synthetic TSP instances of the uniform distribution are as follows:
> > |         | TSP-100 (200)      | TSP-500 (200)       | TSP-1000 (200)      | TSP-5000 (20)      | TSP-10000 (20)     | Average |
> > |------------------------|-------------------|-------------------|-------------------|-------------------|-------------------|-------------|
> > | Method |gap(%)↓, time↓, $n_{bs}$↑     | gap(%)↓, time↓, $n_{bs}$↑    |gap(%)↓, time↓, $n_{bs}$↑   | gap(%)↓, time↓, $n_{bs}$↑     | gap(%)↓, time↓, $n_{bs}$↑    | gap(%)↓        |
> > | nearest neighbor+2-opt | 5.69, 3.7s, 1     | 5.55, 9.4s, 1     | 5.24, 34.9s, 1    | 5.53, 2.7m, 1     | 4.32, 13.5m, 1    | 5.27        |
> > | GELD (Ours)+G          | 1.11, 0.6s, 200   | 2.39, 1.8s, 200   | 2.94, 3.6s, 200  | 7.62, 10.8s, 20  | 9.33, 21.6s, 20   | 4.68        |
> > | GELD (Ours)+BOTH       | **0.06**, 19.2s, 200  | **0.52**, 1.6m, 200   | **0.58**, 3.7m, 200  | **2.77**, 1.8m, 20    | **2.38**, 3.9m, 20    | **1.26**
> >
> > These results demonstrate that GELD outperforms the nearest neighbor+2-opt method in terms of both solution quality and computational efficiency.
> >
> > ---
> >
> > ### Question 1:
> > >In the two-stage training strategy, why was the maximum training size n_max set to 1000? What was the rationale behind this choice, and how did you explore other possible values?
> >
> >
> > ### Response 3.3 (rational behind setting $n_{max}$):
> > The parameter $n_{max}$ was set to 1,000 for the following reasons:
> > 1. A couple of SOTA models [6-7] adopt 1,000 as the maximum node count for large-scale TSP instances.
> > 2. While increasing $n_{max}$ would highly likely improve model generalization, it may also substantially increase the computational cost (as discussed in Appendix B.1). By setting $n_{max}$ = 1,000, we ensured that the second training stage is computationally manageable: completing within 31 hours on our computer.
> >
> > We acknowledge that $n_{max}$ can be set to any value above 100. However, we did not explore alternative values in this study due to the high computational demands of such an investigation.

---

> > > ### Author Response · Authors · 2024-11-24
> > > **Response to Reviewer 6ZAK (Part 3)**
> > >
> > > ### Question 2:
> > > >The Region-Average Linear Attention (RALA) mechanism is a key innovation in the Global-view Encoder (GE). However, it is not clear how the choice of partitioning the nodes into regions (controlled by m_r and m_c) affects the balance between capturing global and local information. Can you provide more insights into this?
> > >
> > > ### Response 3.4 (more insights on the hyperparameter $m$):
> > >
> > > As we discussed in Section 4.1, the hyperparameter $m$ controls the granularity of the regional view:
> > > > "a larger value of $m$ may capture more insights of local regions but increases complexity".
> > >
> > >
> > > Below, we illustrate this dynamic through two extreme cases as follows:
> > > 1. **Case: $m$ = 1.**
> > >    In this configuration, all nodes in the TSP graph are ground into a single region. This setup minimizes computational complexity, as a single regional representation aggregates the global information by averaging all node embeddings. However, the averaging process inherently smooths out local variations, limiting the model's ability to retain detailed node-specific (local) information.
> > > 2. **Case: $m$ is large.**
> > >    When $m$ is sufficiently large, each node is assigned to its own region (while many regions do not contain any node). This configuration maximizes the retention of local information, because each regional representation is directly represented by the individual node embedding. While this setup preserves local details, it significantly increases computational complexity, restricting its applicability to large-scale TSPs.
> > >
> > > These examples underscore the trade-off inherent in the choice of $m$. Smaller values of $m$ favor computational efficiency and emphasize on global information, while larger values prioritize local details at the expense of scalability. Based on practical considerations and as illustrated in Response 3.1, we deem that our choice of $m$=9 is reasonable and leads to an excellent trade-off between performance and inference time.
> > >
> > > ---
> > >
> > > ### Question 3:
> > > >The Local-view Decoder (LD) restricts the candidate set to the k-nearest neighbors. How was the value of k determined, and what is the sensitivity of the model's performance to changes in k?
> > >
> > > ### Response 3.5 (sensitivity analysis on hyperparameter $k_m$):
> > > Thank you for your insightful comment. In our study, we set $k_m$ to 100 to facilitate a fair comparison with baseline models when solving TSPs with fewer than 100 nodes. For such cases, the candidate set of the local-view decoder includes all remaining nodes, which aligns with the configuration of global-view decoder-based baseline models, such as LEHD, BQ, and GD.
> > >
> > > In response to your question, we have now conducted a sensitivity analysis on the model's performance to variations in the parameter $k_m$. The results are as follows:
> > >
> > > |  |  | TSP-100 | TSP-500 | TSP-1000 | TSP-5000 | TSP-10000 ||
> > > |-------|-----------|---------|---------|---------|---------|----------|----------------|
> > > |   Value    | Inference         | gap (%)↓, time↓ | gap (%)↓, time↓ | gap (%)↓, time↓ | gap (%)↓, time↓ | gap (%)↓, time↓ | Average gap (%)↓ |
> > > | 50    | G         | 2.00, 0.4s | 4.02, 1.2s | 5.00, 2.9s | 10.19, 9.2s | 11.03, 18.9s | 6.45 |
> > > |       | BOTH      | 0.17, 18.7s | 0.85, 1.4m | 1.06, 2.9m | 3.20, 1.5m | 3.15, 3.5m | 1.69 |
> > > | 100   | G         | 1.11, 0.6s | 2.39, 1.8s | 2.94, 3.6s | 7.62, 10.8s | 9.33, 21.6s | 4.68 |
> > > |       | BOTH      | 0.06, 19.2s | 0.52, 1.6m | 0.58, 3.7m | 2.77, 1.8m | 2.38, 3.9m | 1.26 |
> > > | 150   | G         | 1.11, 0.7s | 2.28, 2.4s | 2.33, 5.4s | 7.05, 12.1s | 9.52, 24.1s | 4.46 |
> > > |       | BOTH      | 0.06, 21.6s | 0.45, 2.0m | 0.49, 4.3m | 3.11, 2.4m | 2.04, 4.9m | 1.23
> > > ---
> > > As shown, larger values of $k_m$ slightly improve the model performance while taking more time for inference. These findings further highlight the importance of adopting a local-view decoder in our model design.
> > >
> > > ### References
> > > [1] Neural combinatorial optimization with reinforcement learning, ICLR’17 workshop.
> > >
> > > [2] Learning heuristics for the TSP by policy gradient, ICPAIOR’18.
> > >
> > > [3] INViT: A generalizable routing problem solver with invariant nested view transformer, ICML’24.
> > >
> > > [4] UDC: A Unified Neural Divide-and-Conquer Framework for Large-Scale Combinatorial Optimization Problems, NeurIPS’24.
> > >
> > > [5] Attention, learn to solve routing problems! ICLR’19.
> > >
> > > [6] BQ-NCO: Bisimulation quotienting for efficient neural combinatorial optimization, NeurIPS’23.
> > >
> > > [7] Neural Combinatorial Optimization with Heavy Decoder: Toward Large Scale Generalization, NeurIPS’23.

---

> > > > ### Comment · Reviewer_6ZAK · 2024-11-26
> > > > **Thank you for the response**
> > > >
> > > > Thank you for the response. I have carefully read the response and also found the title is misleading with the "All Sizes". For now, I would like to keep my score.

---

> > > > > ### Author Response · Authors · 2024-12-02
> > > > > **Response to Reviewer 6ZAK (Round 2)**
> > > > >
> > > > > ### Comment:
> > > > > >Thank you for the response. I have carefully read the response and also found the title is misleading with the "All Sizes". For now, I would like to keep my score.
> > > > >
> > > > > ### Response 3.6:
> > > > > Thank you for your 2nd-round review comments. We would like to clarify that the term *all sizes* used in the title is an abbreviation for "all scales of problem sizes" and the basis of such usage is led by the following considerations:
> > > > >
> > > > > 1. The term *all sizes* highlights that our proposed unified model is capable of solving TSPs across a wide range of scales, from small-scale instances with 29 nodes to large-scale instances with **744,710 nodes**. To the best of our knowledge, no existing neural TSP solver addresses such a broad spectrum of instance sizes.
> > > > >
> > > > > 2. Our research utilizes real-world datasets such as TSPLIB, National TSPs, and the VLSI dataset, which comprise numerous large-size instances. The excellent performance of GELD on these datasets demonstrates its practical utility and validates its ability to **solve TSPs of all sizes encountered in real-world scenarios**.
> > > > >
> > > > > We hope this clarification addresses your concern regarding the title. Please let us know if you have any further concerns or questions.

---

> > > > > > ### Comment · Reviewer_6ZAK · 2024-12-03
> > > > > > **About All Sizes**
> > > > > >
> > > > > > I still feel the term "All Sizes" is over-claimed. The time complexity of any algorithm is a function of the problem size n. The algorithm demostrated to solve an instance with 744,710 nodes, but it does not mean it can solve instances 1 or 2 magtitude larger. Because the instances are just empirical evaluations of limited samlings from the "instance space". Unless the authors can derive the rigorious time complexity that increases mildly with the size n, I do not think it can be called "All Sizes".

---

> > > > > > > ### Author Response · Authors · 2024-12-03
> > > > > > > **Response to Reviewer 6ZAK (Round 3)**
> > > > > > >
> > > > > > > Thank you so much for the elaboration. Yes, rigorously speaking, although our proposed GELD model is empirically shown as capable of solving a very large TSP instance with 744,710 nodes, we are uncertain about whether it could also effectively solve other instances of sizes 1 or 2 magnitude larger.
> > > > > > >
> > > > > > > More specifically, as given in Figure 1 and discussed in its context, the time complexity of GELD is $\mathcal{O}(nmh)$ for the encoder attention module and $\mathcal{O}(k_m^{\ \ 2}h)$ for the decoder attention module, where $n$ denotes the number of nodes in the underlying instance, $m$ denotes the number of partitioned regions (see Line 276 on Page 6 of our manuscript), and $h$ denotes the dimensionality of the embedding. Although this shows the computational time of GELD will only linearly increase with the problem/instance size, which is significantly more efficient than many models adopting the standard attention mechanism with a time complexity of $\mathcal{O}(n^2h)$ (see Line 47 on Page 1), we may not be 100% certain that GELD will be capable of efficiently solving instances with an extremely large number of nodes (e.g., say a billion nodes).
> > > > > > >
> > > > > > > Hence, per your suggestion, we will change the term *all sizes* to *varying sizes*. If our paper is accepted for publication, we will change the title to the following:
> > > > > > > >From Global Assessment to Local Selection: Efficiently Solving Traveling Salesman Problems of Varying Sizes
> > > > > > >
> > > > > > > Sincerely appreciate all your comments throughout the review and rebuttal journey, which helped us to greatly improve the quality of our manuscript.

---

### Official Review · Reviewer_S4qh · 2024-10-31

**Soundness:** 3
**Presentation:** 3
**Contribution:** 2
**Rating:** 5
**Confidence:** 4

**Summary:**

The paper presents ``GELD``, a neural TSP solver that efficiently handles TSP of various scales and distributions. It features a global encoder and local decoder with a novel attention mechanism, and a two-stage training strategy. The author claims that ``GELD`` outperforms state-of-the-art models in solution quality and speed, offering a promising approach for solving TSP.

**Strengths:**

* The author propose a novel Region-Average Linear Attention (RALA) mechanism within GE which operates with $O(n)$ time-space complexity.
* Using the ``PRC`` post-processing method, ``GELD`` first attempt to solve ultra-large-scale real-world TSP problems.

**Weaknesses:**

* **Weak experimental quantity**: Obviously, as an auto-regressive model, conducting experiments only on TSP is far from enough. I believe, as the author mentioned in the summary section, at least another type of CO problem, such as ``CVRP`` or ``JSSP``, needs to be done.
* **Low novelty**: The proposed ``PRC`` is conceptually similar to ``LCP`` in ``GLOP`` . Methodologically, especially in solving large-scale TSP problems, the two are almost identical, both relying on iterative reconstruction based on random greedy insertion.
* **Poor experimental setup and control**: Firstly, as a well-researched problem, TSP has many publicly available datasets. However, the author didn't test their approach on these datasets, which indirectly prevents comparisons with some of the current state-of-the-art TSP neural solvers, such as ``DIFUSCO`` and ``T2T``.  Secondly, when referencing ``BQ-NCO``, only their greedy approach was cited, not their beam-search, while ``GELD`` uses beam-search.

**Questions:**

* Please explain the three points mentioned in the ``Weaknesses`` section.
* Traditional solvers like ``LKH`` have already performed exceptionally well on TSP problems, and works like ``NeuroLKH`` further demonstrate that ``LKH`` can achieve high-quality solutions for large-scale TSP instances in very short times. So what are the advantages of Neural-TSP-Solvers like the one in this paper?
* Question regarding GELD's experimental data: I retested the uniform data test set locally using ``LKH``, but the results I obtained differ significantly from those in the repository. Below are my results. It should be noted that I used 20 threads for the solution, so I also divided the reported time in the paper by 20 accordingly.

|           |  Author Settings   | Author Results | Reviewer Settings | Reviewer Results |
|:---------:|:------------------:|:--------------:|:-----------------:|:----------------:|
|  TSP-100  | 20K iters, 10 runs |   7.8693, 8s   | 500 iters, 1 runs |    7.8694, 6s    |
|  TSP-500  | 20K iters, 10 runs | 16.5601, 666s  | 500 iters, 1 runs |   16.5532, 47s   |
| TSP-1000  | 20K iters, 10 runs | 23.2215, 2736s | 500 iters, 1 runs |   23.1560, 74s   |
| TSP-5000  | 20K iters, 1 runs  | 50.9830, 306s  | 500 iters, 1 runs |   50.9860, 72s   |
| TSP-10000 | 20K iters, 1 runs  | 73.1436, 5616s | 500 iters, 1 runs |  71.8154, 261s   |

---

> ### Author Response · Authors · 2024-11-24
> **Response to Reviewer S4qh (Part 1)**
>
> Thank you for your review comments and providing us the insightful and critical feedback. Here are our detailed responses to your comments. We are more than happy to discuss any further concerns or questions that you may have.
>
> ---
>
> ### Weakness 1:
> >Weak experimental quantity: Obviously, as an auto-regressive model, conducting experiments only on TSP is far from enough. I believe, as the author mentioned in the summary section, at least another type of CO problem, such as CVRP or JSSP, needs to be done.
>
> ### Response 2.1 (weak experimental quantity):
> Our study focuses exclusively on TSP to ensure **the depth of our research** (i.e., using a unified model that both functions as a standalone solver and as an improvement method to solve various sorts of TSPs of all sizes), which coincides with the scope of recent studies published in top-tier conferences such as ICLR, ICML, and NeurIPS (see Response 1.1 to Reviewer Wg4P for details). **Therefore, we strongly believe that given the depth of our research, comprehensive experimentation (on various sorts of TSPs of all sizes), and SOTA results achieved, our revised manuscript deserves an impartial consideration of being accepted by the flagship ML conference ICLR.**
>
> We acknowledge that using a unified model to effectively solve COPs (not only TSPs) of all sizes while efficiently enhancing solution quality is an exciting and worthwhile direction for future work. However, these problems involve greater complexity and distinct requirements, making it **challenging to comprehensively address them alongside TSP in a single study**.
>
> Nonetheless, we are optimistic about the (future) application of GELD on other COPs because prior studies have shown that models performing well on TSP can achieve excellent results on other COPs (e.g., CVRP [1], JSSP [2], and Maximum Independent Set [3]). Importantly, GELD does not rely on operations (often) specifically tailored to TSP, such as heatmap+2-opt or Monte Carlo Tree Search Sampling [4].
>
> Because we had conducted extensive experiments on TSPs of various sizes (ranging from 29 to 744,710 nodes) and distributions (including four synthetic and real-world patterns) to demonstrate the robustness and effectiveness of our proposed GELD model, we respectfully disagree with the characterization of the experimental quality as "weak".
>
> ---
>
> ### Weakness 2:
> >Low novelty: The proposed PRC is conceptually similar to LCP in GLOP. Methodologically, especially in solving large-scale TSP problems, the two are almost identical, both relying on iterative reconstruction based on random greedy insertion.
>
> ### Response 2.2 (low novelty):
> We would like to clarify that one key innovation of our work lies in **the enhancement of the effectiveness of re-construction by increasing the diversification of model inputs, not the concept of PRC itself** (see Sections 3.4 and 4.1).
>
> Methodologically, our proposed model differs from GLOP [5] in two significant aspects as follows:
>
> 1. **Dual Functionality**
>    As stated in Section 2.2, *"To the best of our knowledge, there does not exist a unified approach capable of both producing and improving TSP solutions"*. Unlike GLOP [5], which functions solely as a post-processing method relying on initial solutions generated via random greedy insertion, our model functions both as a **neural construction model** for independently generating TSP solutions and as a **post-processing method** for improving initial solutions produced by various models.
>
> 2. **Unified Parameterization**
>    GLOP requires multiple pre-trained models (e.g., trained separately for TSP-20, TSP-50, and TSP-100) while our model employs a single set of pre-trained parameters. **This unified model reduces the reliance on scale-specific training and enhances the model's practical applicability, a primary goal of our research and a key focus in the field**.
>
> We emphasize that the use of random greedy insertion algorithm in our study is intended to demonstrate the effectiveness of our model as a post-processing method, not to establish dependency on it. Unlike GLOP, which inherently depends on the random greedy insertion algorithm, our model demonstrates significant improvements over initial solutions across eight baselines (see Table 3).
>
> Therefore, we believe the assessment of low novelty may stem from a misunderstanding of our contributions. While conceptual similarities between our work and GLOP exist, **ours addresses critical challenges in the field by proposing innovative architectural solutions and achieving improved scalability and versatility**. Therefore, we respectfully disagree with the characterization of our work as lacking novelty.
>
> ---

---

> > ### Author Response · Authors · 2024-11-24
> > **Response to Reviewer S4qh (Part 2)**
> >
> > ### Weakness 3:
> > >Poor experimental setup and control: Firstly, as a well-researched problem, TSP has many publicly available datasets. However, the author didn't test their approach on these datasets, which indirectly prevents comparisons with some of the current state-of-the-art TSP neural solvers, such as DIFUSCO and T2T. Secondly, when referencing BQ-NCO, only their greedy approach was cited, not their beam-search, while GELD uses beam-search.
> >
> > ### Response 2.3 (poor experimental setup and control):
> > We address your concerns as follows.
> >
> > ##### 1. Experimental Datasets
> > We did design the experiments for wide coverage and in-depth analysis. Specifically, we followed a well-recognized, commonly adopted approach as presented in [6] and utilized two widely recognized types of datasets for evaluation:
> > - **Synthetic Data**: The MSVDRP dataset from [6], encompassing TSPs of 5 sizes (TSP-{100, 500, 1000, 5000, 10000}) and 4 distribution patterns (uniform, clustered, explosion, and implosion). Each of these 5×4=20 subsets contains 200 instances (for TSP-{100, 500, 1000}) or 20 instances (for TSP-{5000, 1000}).
> > - **Real-world Data**: Benchmark datasets such as TSPLIB (77 TSP instances) and World TSP (27 National TSP instances and 4 extremely large TSP instances).
> >
> > The synthetic and real-world datasets are commonly used to evaluate the performance of neural TSP solvers, including the models DIFUSCO and T2T, mentioned in your comment, and many other SOTA studies [5-7]. Therefore, we are confused about the claim that our work neglects many publicly available datasets.
> >
> > ##### 2. Comparison Methods
> > - **DIFUSCO**: Regarding the comparison with DIFUSCO, please refer to Response 1.4 to Reviewer Wg4P for details. In essence, GELD outperforms DIFUSCO in terms of both solution quality and computational efficiency.
> > - **T2T**: We encountered technical difficulties when attempting to execute the source code of T2T, specifically the error: `xt.grad is not None`. Unfortunately, these issues prevented us from including T2T in our baseline comparisons.
> >
> > ##### 3. BQ-NCO Setting
> > We focused on the greedy approach of BQ-NCO [7] for the following two reasons:
> > 1. The beam-search implementation of BQ-NCO is computationally impractical for large-scale TSPs (e.g., over 5,000 nodes ) due to GPU memory constraints and excessive inference time.
> > 2. Our setup follows [6], which reports only the greedy search results for BQ-NCO.
> >
> > Importantly, **we did include greedy results for our model** (see Response 1.4 to Reviewer Wg4P for more details). In comparisons under greedy search, our model consistently demonstrated superior efficiency and performance over BQ-NCO.
> >
> > In summary, we respectfully disagree with the assertion that our experimental setups and control are poor. We believe our experimental setups are justified and align well with most established practices in the field.
> >
> > ---
> >
> > ### Question 1:
> > >Traditional solvers like LKH have already performed exceptionally well on TSP problems, and works like NeuroLKH further demonstrate that LKH can achieve high-quality solutions for large-scale TSP instances in very short times. So what are the advantages of Neural-TSP-Solvers like the one in this paper?
> >
> > ### Response 2.4 (advantages of neural TSP solvers over NeuroLKH):
> > Thank you for your question. We identify the following two key advantages of neural TSP solvers compared to NeuroLKH:
> >
> > 1. **Reduced Dependence on Expert Knowledge:**
> >    NeuroLKH is an improvement-based approach that heavily relies on human-designed operators and advanced TSP solvers (i.e., LKH). In contrast, neural TSP solvers aim to develop purely learning-based solvers for TSP [1, 2, 6, 7]. This approach reduces the dependence of TSP solvers on expert-designed components and broadens the applicability of these methods to other COPs.
> >
> > 2. **Enhanced Inference Speed:**
> >    NeuroLKH's runtime includes both the execution of its sparse graph network model and the computational overhead of LKH, which can be substantial for large-scale TSPs. In contrast, the runtime of construction-based neural TSP solvers only involves the runtime of the model itself. Additionally, neural TSP solvers benefit from highly parallelized GPU computing, enabling faster and more efficient problem-solving capabilities.
> >
> > ---

---

> > > ### Author Response · Authors · 2024-11-24
> > > **Response to Reviewer S4qh (Part 3)**
> > >
> > > ### Question 2:
> > > >Question regarding GELD's experimental data: I retested the uniform data test set locally using LKH, but the results I obtained differ significantly from those in the repository. Below are my results. It should be noted that I used 20 threads for the solution, so I also divided the reported time in the paper by 20 accordingly.
> > >
> > > ### Response 2.5 (experimental data):
> > > Thank you for your insightful question. As outlined in Appendix B.2 of the submitted manuscript, the synthetic datasets were generated following [6]. Specifically, to minimize environmental biases, we directly utilized the publicly available MSVDRP dataset (TSP-{100, 1000, 5000, 10000}) from [6], which includes synthetic instances and their corresponding solutions generated using LKH. Additionally, the TSP-500 dataset was created using the instance and solution generation code provided in the same study (the datasets and code available via [8]).
> > >
> > > After receiving your question, we verified the experimental settings and re-conducted the relevant experiments. The results of the re-conducted experiments remain the same as reported in our submitted manuscript.
> > >
> > > The discrepancies in your LKH results, particularly for TSP-10000, may be attributed to differences in the versions of the LKH solver. Specifically, the code referenced in [6] employs ELKAI, a Python-based implementation of LKH, which can produce results that slightly differ from those generated by other versions of the solver. For further clarification, we suggest contacting the authors of [6].
> > >
> > > It is also important to highlight that our evaluation metric focuses on the **performance gap relative to the LKH results** (see Appendix B.4). Therefore, variations in LKH outputs do not affect the primary conclusion that GELD consistently outperforms seven state-of-the-art baseline models.
> > >
> > > ---
> > >
> > > ### References
> > > [1] Neural combinatorial optimization with heavy decoder: Toward large scale generalization, NeurIPS’24.
> > >
> > > [2] Self-improvement for neural combinatorial optimization: Sample without replacement, but improvement, TMLR'24.
> > >
> > > [3] DISCO: Efficient diffusion solver for large-scale combinatorial optimization problems, arXiv'24.
> > >
> > > [4] DIFUSCO: Graph-based diffusion solvers for combinatorial optimization, NeurIPS’23.
> > >
> > > [5] GLOP: Learning global partition and local construction for solving large-scale routing problems in real-time, AAAI’24.
> > >
> > > [6] INViT: A generalizable routing problem solver with invariant nested view transformer, ICML’24.
> > >
> > > [7] BQ-NCO: Bisimulation quotienting for efficient neural combinatorial optimization, NeurIPS’23.
> > >
> > > [8] https://github.com/Kasumigaoka-Utaha/INViT

---

> > > > ### Comment · Reviewer_S4qh · 2024-11-25
> > > > **Thank you for the authors' responses.**
> > > >
> > > > 1. The datasets I am referring to are the following, all of which are publicly available and unified
> > > > * ``DIMES``: https://github.com/DIMESTeam/DIMES/tree/main/TSP/data
> > > > * ``Att-GCN`` and ``DIFUSCO``: https://github.com/Spider-scnu/TSP/tree/master/MCTS
> > > > * ``T2TCO``: https://github.com/Thinklab-SJTU/T2TCO/tree/main/data/tsp
> > > > 2. I hope you can provide the solution length of ``LKH`` and neural models. Since the solver you used is not an exact solution, so only reporting the gap is unfair to other articles that you didn't cite. Alternatively, you can use Concorde as the baseline, and the corresponding dataset can be accessed through this link: https://github.com/Thinklab-SJTU/T2TCO/tree/main/data/tsp
> > > > 3. I have the same question as the reviewer ``Wg4P``. ``The authors have mentioned that the biggest contribution/novelty is addressing the lack of a unified method for both producing and improving TSP solutions. Is it necessary to unify the two? Do you have any insights on this?``
> > > > 4. I do not fully agree with the statement that ``BQ-NCO`` should not be excerpted due to reasons such as ``OOM`` and computing speed. The author can point out ``OOM`` in the paper instead of choosing to ignore it.

---

> ### Author Response · Authors · 2024-12-02
> **Response to Reviewer S4qh (Round 2, Part 1)**
>
> Thank you for your 2nd-round review comments. Here are our detailed responses to your comments.
>
> ### Comment 1:
> >The datasets I am referring to are the following, all of which are publicly available and unified:
> DIMES: https://github.com/DIMESTeam/DIMES/tree/main/TSP/data
> Att-GCN and : https://github.com/Spider-scnu/TSP/tree/master/MCTS DIFUSCO
> T2TCO: https://github.com/Thinklab-SJTU/T2TCO/tree/main/data/tsp
>
> ### Response 2.6:
>
> Thank you for providing a list of publicly available synthetic datasets. While the provided datasets are publicly available, all of them are limited to uniformly distributed TSP instances. In contrast, we used the publicly available MSVDRP dataset (ICML'24), which includes instances from four distribution patterns: Uniform, Clustered, Explosion, and Implosion, as stated in our previous response (Response 2.3, Round 1):
> >**Synthetic Data**: The MSVDRP dataset from [1], encompassing TSPs of 5 sizes (TSP-{100, 500, 1000, 5000, 10000}) and 4 distribution patterns (uniform, clustered, explosion, and implosion). Each of these 5×4=20 subsets contains 200 instances (for TSP-{100, 500, 1000}) or 20 instances (for TSP-{5000, 1000}).
>
> **We believe the datasets encompassing these four distribution patterns better reflect the variety and complexity encountered in real-world scenarios than those only having one distribution pattern**. Furthermore, we conducted experiments on the real-world TSPLIB dataset, which is also used in the studies you referenced [2–3]. Therefore, we consider our choice of datasets as reasonable and representative.
>
> ---
>
> ### Comment 2:
> >I hope you can provide the solution length of LKH and neural models. Since the solver you used is not an exact solution, so only reporting the gap is unfair to other articles that you didn't cite. Alternatively, you can use Concorde as the baseline, and the corresponding dataset can be accessed through this link: https://github.com/Thinklab-SJTU/T2TCO/tree/main/data/tsp
>
> ### Response 2.7:
> Thank you for your comments. We have now included the LKH results on the synthetic dataset in Table 6 of the revised manuscript. The solution lengths of the neural models can be directly derived from the reported gaps based on the LKH results.
>
> Regarding your statement that "Since the solver you used is not an exact solution, so only reporting the gap is unfair to other articles that you didn't cite", we respectfully disagree on this for the following reasons:
>
> 1. Use of Exact Solvers
>
>    While exact solvers such as Concorde provide optimal solutions, they are computationally infeasible for large-scale TSPs (e.g., TSP-10000). Moreover, even with exact solvers, the resulting gaps reported in one study could not directly be used in other studies, because differences in (synthetic) datasets and solver configurations can lead to inconsistent benchmarks. We believe that a fair comparison can only be made if all baseline models are evaluated under the same conditions, as we have done in all experiments in this work.
>
> 2. Comprehensive Evaluation Metrics
>
>    Evaluating model performance requires the assessment of both solution quality and inference speed. Focusing exclusively on solution quality may neglect critical practical aspects such as inference speed, which is particularly vital in real-world applications like smart logistics hubs and automated ports and terminals. Therefore, we report results for all baseline models under consistent experimental settings on our hardware. Directly comparing solution quality across models without running the respective code to get the inference time would fall short in the evaluations of reliability and comprehensiveness.
>
> Nonetheless, per your suggestion, we have now conducted additional experiments using the datasets accessed through the links you provided (see Appendix B.8 of the revised manuscript).

---

> > ### Author Response · Authors · 2024-12-02
> > **Response to Reviewer S4qh (Round 2, Part 2)**
> >
> > The performance of GELD on the dataset in the link https://github.com/Thinklab-SJTU/T2TCO/tree/main/data/tsp is as follows:
> >
> > | **Method**  | **TSP-50 (1280)** |             |                            | **TSP-100 (1280)** |             |                            | **TSP-500 (128)** |             |                            | **TSP-1000 (128)** |             |                            |
> > | ----------- | ----------------- | ----------- | -------------------------- | ------------------ | ----------- | -------------------------- | ----------------- | ----------- | -------------------------- | ------------------ | ----------- | -------------------------- |
> > |             | **Length↓**       | **gap(%)↓** | **time↓, $n_{bs}$↑** | **Length↓**        | **gap(%)↓** | **time↓, $n_{bs}$↑** | **Length↓**       | **gap(%)↓** | **time↓, $n_{bs}$↑** | **Length↓**        | **gap(%)↓** | **time↓, $n_{bs}$↑** |
> > | Concorde    | 5.6876            | -           | -                          | 7.7559             | -           | -                          | 16.5458           | -           | -                          | 23.1181            | -           | -                          |
> > | GELD + G    | 5.7467            | 1.04        | 0.6s, 1280                 | 7.8377             | 1.05        | 1.2s, 1280                 | 16.9601           | 2.50        | 1.2s, 128                  | 23.9285            | 3.50        | 2.4s, 128                  |
> > | GELD + BOTH | 5.6889            | 0.02        | 35.4s, 1280                | 7.7604             | 0.06        | 1.6m, 1280                 | 16.6334           | 0.53        | 1.1m, 128                  | 23.3209            | 0.88        | 2.2m, 128                  |
> >
> > The performance of GELD on the dataset in the link https://github.com/DIMESTeam/DIMES/tree/main/TSP/data is as follows:
> >
> > | **Method**  | **TSP-100 (10000)** |             |                            | **TSP-500 (128)** |             |                            | **TSP-1000 (128)** |             |                            | **TSP-10000 (16)** |             |                            |
> > | ----------- | ------------------- | ----------- | -------------------------- | ----------------- | ----------- | -------------------------- | ------------------ | ----------- | -------------------------- | ------------------ | ----------- | -------------------------- |
> > |             | **Length↓**         | **gap(%)↓** | **time↓, $n_{bs}$↑** | **Length↓**       | **gap(%)↓** | **time↓, $n_{bs}$↑** | **Length↓**        | **gap(%)↓** | **time↓, $n_{bs}$↑** | **Length↓**        | **gap(%)↓** | **time↓, $n_{bs}$↑** |
> > | Concorde    | 7.7645              | -           | -                          | 16.5836           | -           | -                          | 23.2268            | -           | -                          | 71.77              | -           | -                          |
> > | GELD + G    | 7.8419              | 1.00        | 9.1s, 1000                | 16.9601           | 2.27        | 1.2s, 128                  | 23.9285            | 3.02        | 2.4s, 128                  | 79.7566            | 11.13       | 18.2s, 16                   |
> > | GELD + BOTH | 7.7659              | 0.02        | 11.9m, 1000               | 16.6334           | 0.30        | 1.1m, 128                  | 23.3209            | 0.41        | 2.2m, 128                  | 75.1468            | 4.71        | 3.2m, 16                  |

---

> > > ### Author Response · Authors · 2024-12-02
> > > **Response to Reviewer S4qh (Round 2, Part 3)**
> > >
> > > The performance of GELD on the dataset in the link  https://github.com/Spider-scnu/TSP/tree/master/MCTS is as follows:
> > > | **Method**      | **TSP-20 (10000)** |             |                            | **TSP-50 (10000)** |             |                            | **TSP-100 (10000)** |             |                            | **TSP-200 (128)** |             |                            | **TSP-500 (128)** |             |                            | **TSP-1000 (128)** |             |                            | **TSP-10000 (16)** |             |                            |
> > > | --------------- | ------------------ | ----------- | -------------------------- | ------------------ | ----------- | -------------------------- | ------------------- | ----------- | -------------------------- | ----------------- | ----------- | -------------------------- | ----------------- | ----------- | -------------------------- | ------------------ | ----------- | -------------------------- | ------------------ | ----------- | -------------------------- |
> > > |                 | **Length↓**        | **gap(%)↓** | **time↓, $n_{bs}$↑** | **Length↓**        | **gap(%)↓** | **time↓, $n_{bs}$↑** | **Length↓**         | **gap(%)↓** | **time↓, $n_{bs}$↑** | **Length↓**       | **gap(%)↓** | **time↓, $n_{bs}$↑** | **Length↓**       | **gap(%)↓** | **time↓, $n_{bs}$↑** | **Length↓**        | **gap(%)↓** | **time↓, $n_{bs}$↑** | **Length↓**        | **gap(%)↓** | **time↓, $n_{bs}$↑** |
> > > | **Concorde**    | 3.8306             | -           | -                          | 5.6918             | -           | -                          | 7.7645              |             |                            | 10.7280           |             |                            | 16.5836           |             |                            | 23.2268            |             |                            | 71.7700            |             |                            |
> > > | **GELD + G**    | 3.8838             | 1.39        | 0.6s, 10000                | 5.7502             | 1.03        | 3s, 2500                   | 7.8419              | 1.00        | 10.8s, 1250                | 10.9159           | 1.75        | 0.6s, 128                  | 16.9601           | 2.27        | 1.2s, 128                  | 23.9285            | 3.02        | 18.3s, 128                  | 79.7566            | 11.13       | 18.6s, 16                  |
> > > | **GELD + BOTH** | 3.8306             | 0.00        | 1.8m, 10000                | 5.6919             | 0.00        | 5.0m, 2500                | 7.7659              | 0.02        | 12.3m, 1250               | 10.7485           | 0.19        | 0.42m, 128                 | 16.6334           | 0.30        | 1.1m, 128                 | 23.3209            | 0.41        | 3.20m, 128                 | 75.1468            | 4.71        | 3.2m, 16                  |
> > >
> > > These results demonstrate that our proposed GELD achieves excellent performance across all datasets.
> > >
> > > ---
> > >
> > > ### Comment 3:
> > > >I have the same question as the reviewer Wg4P. The authors have mentioned that the biggest contribution/novelty is addressing the lack of a unified method for both producing and improving TSP solutions. Is it necessary to unify the two? Do you have any insights on this?
> > >
> > > ### Response 2.8:
> > >
> > > Please refer to Response 1.6 (Round 2) to Reviewer Wg4P for details.

---

> > > > ### Author Response · Authors · 2024-12-02
> > > > **Response to Reviewer S4qh (Round 2, Part 4)**
> > > >
> > > > ### Comment 4:
> > > > >I do not fully agree with the statement that BQ-NCO should not be excerpted due to reasons such as OOM and computing speed. The author can point out OOM in the paper instead of choosing to ignore it.
> > > >
> > > > ### Response 2.9:
> > > >
> > > > Thank you for your comment. We attempted to use BQ with beam search (Beam = 16, the default setting) to solve TSP instances on the synthetic datasets. However, we encountered an OOM issue on a 24GB GPU when the node scale reached 5,000. Furthermore, BQ with beam search required 2.5 hours to solve TSP-1000 instances (with a batch size of 5). In contrast, our proposed GELD method solved TSP-1000 instances in several minutes with a batch size of 200.
> > > >
> > > > To ensure a fair comparison of computational efficiency, we reported only the results of BQ using the greedy strategy. This approach is consistent with recent studies, including those published in ICML'24 [1] and submitted to ICLR'25 [4].
> > > >
> > > > Nonetheless, in response to your feedback, we have now clarified this rationale in the revised manuscript (see Appendix B.3):
> > > >
> > > > >Furthermore, for a fair comparison in terms of computational efficiency, we report results only for the two baseline models (LEHD and BQ) combined with the greedy search strategy.
> > > >
> > > > ---
> > > >
> > > > ### References
> > > >
> > > > [1] INViT: A generalizable routing problem solver with invariant nested view transformer, ICML'24.
> > > >
> > > > [2] DIFUSCO: Graph-based diffusion solvers for combinatorial optimization, NeurIPS'23.
> > > >
> > > > [3] T2T: From Distribution Learning in Training to Gradient Search in Testing for Combinatorial Optimization, NeurIPS'23.
> > > >
> > > > [4] DISCO: Efficient diffusion solver for large-scale combinatorial optimization problems, submitted to ICLR'25.

---

> > > > > ### Comment · Reviewer_S4qh · 2024-12-02
> > > > > **Thank you for the author's response.**
> > > > >
> > > > > Thank you for the author's response. I agree with the author's statement that 'Evaluating model performance requires the assessment of both solution quality and inference speed'. From the current methods available in the ML4TSP community, the advantage of GELD lies in solving large-scale TSP problems with TSP1000 and above, which can obtain good solutions in a relatively fast time. In view of this, I have decided to increase my score to 5.

---

> > > > > > ### Author Response · Authors · 2024-12-03
> > > > > > **Official Comment by Authors**
> > > > > >
> > > > > > Thank you very much for raising the score!  We are delighted that we have addressed all your concerns.

---

### Official Review · Reviewer_Wg4P · 2024-11-01

**Soundness:** 3
**Presentation:** 2
**Contribution:** 2
**Rating:** 5
**Confidence:** 4

**Summary:**

Authors propose a neural solver GELD which integrates a lightweight Global-view Encoder with a heavyweight Local-view Decoder to enrich embedding representation
while accelerating the decision-making process. Experimental results show that GELD is able to significantly improve the  solution quality of both small-scale and large-scale TSP instances with affordable computing time.

**Strengths:**

1. This paper is easy to follow.
2. The Region-Average Linear Attention Structure seems interesting and could be applied in other applications.

**Weaknesses:**

1. The motivation is not clear. Authors mention that Divide&Conquer may fail to provide valuable insights for solving other COPs such as JSSP, but the results of other problems are missing.
2. The contribution 1 is not rigorous. Heatmap-based methods such as Att-GCN and D&R methods Select and Optimize[1] also use the same set of pre-trained model parameters for all the  TSP scales.
3. Some important points are missing. GELD seems to be an iterative framework, and the local part of the solution needs to be reconstructed for each iteration. The detailed illustration of the working process is missing.
4. The comparison methods in experiments are not complete. Heatmap-based approaches(Att-GCN[2], DIFUSCO[3], etc.) and the recent work ICAM[4] should be included. Furthermore, it seems unfair to combine GELD with different search strategies such as BS and PRC, while only the greedy search results are listed for other baselines

**Questions:**

See weekness.

Reference:

1. Hanni Cheng, Haosi Zheng, Ya Cong, Weihao Jiang, and Shiliang Pu. Select and optimize: Learning to aolve large-scale tsp instances. AISTATS2023.

2. Zhang-Hua Fu, Kai-Bin Qiu, and Hongyuan Zha. Generalize a small pre-trained model to arbitrarily
large tsp instances. AAAI2021.

3. Zhiqing Sun and Yiming Yang. DIFUSCO: Graph-based diffusion solvers for combinatorial optimization. NeurIPS2023.

4. Changliang Zhou, Xi Lin, Zhenkun Wang, Xialiang Tong, Mingxuan Yuan, and Qingfu Zhang. Instanceconditioned adaptation for large-scale generalization of neural combinatorial optimization. arXiv preprint
arXiv:2405.01906, 2024

---

> ### Author Response · Authors · 2024-11-24
> **Response to Reviewer Wg4P (Part 1)**
>
> Thank you for your review comments and providing us the insightful and critical feedback. Here are our detailed responses to your comments. We are more than happy to discuss any further concerns or questions that you may have.
>
> ---
>
> ### Weakness 1:
>
> > The motivation is not clear. Authors mention that Destroy&Repair may fail to provide valuable insights for solving other COPs such as JSSP, but the results of other problems are missing.
>
> ### Response 1.1 (unclear motivation):
> First of all, we would like to clarify that our manuscript did not claim that the Destroy & Repair (D&R) strategy may fail to provide insights for solving other combinatorial optimization problems (COPs). **Instead, the discussion in the Introduction section highlights the limitations of the Divide & Conquer (D&C) approach**. The D&C strategy fundamentally differs from the D&R-based strategies such as re-construction which we had discussed in Section 3.4 of our manuscript.
>
> Regarding the absence of results on other COPs, our study focuses exclusively on the Traveling Salesman Problem (TSP) to ensure research depth. Specifically, we fill the corresponding research gap by developing a unified model that functions both as a standalone solver and as an improvement method to solve TSPs of all sizes. Furthermore, we conduct extensive experiments on 20 synthetic datasets (across 5 scales and comprising 4 distribution patterns) and 2 real-world datasets (77 instances from TSPLIB and 27 National TSP instances and 4 extremely large TSP instances from World TSP) with 7 state-of-the-art (SOTA) baseline models to demonstrate the effectiveness of our model. Our in-depth study coincides with the scope of a number of recent studies published in top-tier conferences, such as ICLR [1], NeurIPS [2], ICML [3], AAAI [4–7], SIGKDD [8], and AISTATS [9], and some submitted to ICLR'25 [10-12]. These works also focus exclusively on TSP without conducting experiments on other COPs. **Therefore, we strongly believe that given the depth of our research, comprehensive experimentation (on various sorts of TSPs of all sizes), and SOTA results achieved, our revised manuscript deserves an impartial consideration of being accepted by the flagship ML conference ICLR.**
>
> While our current study only presents the results on TSP, we are optimistic about the (future) application of GELD on other COPs because prior studies have shown that models performing well on TSP can achieve excellent results on other COPs (e.g., CVRP [13] and JSSP [14]). Importantly, GELD does not rely on operations (often) specifically tailored to TSP, such as heatmap+2-opt or Monte Carlo Tree Search Sampling [15].
>
> To sum it up, we believe the manuscript’s motivation is clear and appropriately presented. It is justified for us to exclusively focus on TSP among many other COPs by providing in-depth technical contributions and comprehensive experimental results on various types of TSPs of all sizes.
>
> ---
>
> ### Weakness 2:
> >The contribution 1 is not rigorous. Heatmap-based methods such as Att-GCN and D&R methods Select and Optimize also use the same set of pre-trained model parameters for all the TSP scales.
>
> ### Response 1.2 (contribution 1 not rigorous ):
> We would like to emphasize that **GELD fundamentally differs from the models you referenced**. Specifically:
> - **Comparison to Att-GCN [7] and Select-and-Optimize [9]**:
>   - Both Att-GCN and Select-and-Optimize leverage **advanced heuristic algorithms**, namely Monte Carlo Tree Search in Att-GCN and refined POPMUSIC and Lin-Kernighan algorithm in Select-and-Optimize, and only function as **improvement methods** to iteratively refine initial solutions.
>   - In contrast, GELD is a TSP solver that **can learn from scratch**, capable of directly generating high-quality solutions as a **neural construction model**, while additionally functioning as an **improvement method** to optimize initial solutions.
>
> In response to your feedback, we have revised the first contribution for greater clarity and precision as follows:
>
> > "To the best of our knowledge, GELD is the first unified model with a single set of pre-trained parameters that effectively solves TSPs of all sizes while efficiently enhancing solution quality."
>
> We believe this refinement more accurately highlights GELD's novelty and distinctiveness compared to existing methods.
>
> ---

---

> > ### Author Response · Authors · 2024-11-24
> > **Response to Reviewer Wg4P (Part 2)**
> >
> > ### Weakness 3:
> > >Some important points are missing. GELD seems to be an iterative framework, and the local part of the solution needs to be reconstructed for each iteration. The detailed illustration of the working process is missing.
> >
> >
> > ### Response 1.3 (missing important points):
> > GELD is not merely an iterative framework but **a comprehensive approach** to addressing the lack of a unified method for both producing and improving TSP solutions. As claimed in Section 2.1, *"To the best of our knowledge, there does not exist a unified approach capable of both producing and improving TSP solutions"*. GELD fills this gap by serving as both a standalone TSP solver and a powerful post-processing method for iteratively improving initial solutions.
> >
> > Regarding your inquiry on the detailed illustration of the re-construction process: GELD randomly selects sub-solutions, reintegrates their node features into the model, and generates new sub-solutions using a greedy strategy. If these new sub-solutions are of higher quality, they replace the current ones. **This process had been thoroughly detailed in Section 3.4, with additional mathematical explanations provided in Appendix A.1**.
> >
> > To sum it up, we believe that the important points mentioned by you had been clearly and adequately presented in the submitted manuscript.
> >
> > ---
> >
> > ### Weakness 4:
> >
> > >The comparison methods in experiments are not complete. Heatmap-based approaches(Att-GCN, DIFUSC, etc.) and the recent work ICAM should be included. Furthermore, it seems unfair to combine GELD with different search strategies such as BS and PRC, while only the greedy search results are listed for other baselines
> >
> > ### Response 1.4 (incomplete comparison methods & unfair experimental setting):
> > We address your concerns regarding the completeness of the comparison methods and the fairness of experimental setups as follows.
> >
> > ##### 1. Completeness of Comparison Methods
> > We respectfully disagree with your assertion that the comparison methods are incomplete. Our study includes **seven SOTA baseline models: Omni-TSP (ICML’23), LEHD (NeurIPS’23), BQ-NCO (NeurIPS’23), ELG (IJCAI’24), INViT-3V (ICML’24), GD (TMLR’24), and UDC (NeurIPS’24)**. These models represent cutting-edge advancements in the field and provide a comprehensive foundation for comparative analysis.
> >
> > Regarding the omission of DIFUSCO and Att-GCN, these methods were excluded because prior studies [16-18] have demonstrated that our selected baseline models such as BQ, ELG, and UDC outperform them. ICAM was excluded due to the absence of publicly available code and potentially insufficient implementation details in its original paper. Furthermore, ICAM’s reported performance on TSPs with over 2,000 nodes lags behind models like LEHD, BQ, and INViT-3V. Because GELD surpasses these baselines, it is reasonable to infer that GELD would also outperform ICAM under similar conditions. Additionally, UDC [18], which employs ICAM as its conquering policy model, is directly outperformed by GELD in our experiments, further supporting our assertion.
> >
> > Nonetheless, in response to your comment, we have now included additional comparative experiments with DIFUSCO [15] and the heuristic algorithm of nearest neighbor + 2-opt (see Appendix B.6 of the revised manuscript for more details). The comparison results on the synthetic TSP instances of the uniform distribution are as follows:
> > |         | TSP-100 (200)      | TSP-500 (200)       | TSP-1000 (200)      | TSP-5000 (20)      | TSP-10000 (20)     | Average |
> > |------------------------|-------------------|-------------------|-------------------|-------------------|-------------------|-------------|
> > | Method |gap(%)↓, time↓, $n_{bs}$↑     | gap(%)↓, time↓, $n_{bs}$↑    |gap(%)↓, time↓, $n_{bs}$↑    | gap(%)↓, time↓, $n_{bs}$↑     | gap(%)↓, time↓, $n_{bs}$↑    | gap(%)↓        |
> > | nearest neighbor+2-opt | 5.69, 3.7s, 1     | 5.55, 9.4s, 1     | 5.24, 34.9s, 1    | 5.53, 2.7m, 1     | 4.32, 13.5m, 1    | 5.27        |
> > | DIFUSCO+S+2-opt        | 0.06, 2.0m, 1     | 3.95, 22.1m, 1   | 3.33, 1.6h, 1    | 6.54, 37.8m, 1   | 4.72, 2.1h, 1    | 3.72        |
> > | GELD (Ours)+G          | 1.11, 0.6s, 200   | 2.39, 1.8s, 200   | 2.94, 3.6s, 200  | 7.62, 10.8s, 20  | 9.33, 21.6s, 20   | 4.68        |
> > | GELD (Ours)+BOTH       | **0.06**, 19.2s, 200  | **0.52**, 1.6m, 200   | **0.58**, 3.7m, 200  | **2.77**, 1.8m, 20    | **2.38**, 3.9m, 20    | **1.26**        |
> >
> > Notably, the performance of DIFUSCO is heavily dependent on the iterative optimization process of 2-opt that (often) specifically tailored to TSP, while our method does not. More importantly, as shown in this table, GELD outperforms the two compared methods in terms of both solution quality and computational efficiency.

---

> > > ### Author Response · Authors · 2024-11-24
> > > **Response to Reviewer Wg4P (Part 3)**
> > >
> > > ##### 2. Fairness of Comparisons
> > > We also respectfully disagree with the claim that our comparisons are unfair. In all experiments (see all tables presented in the Experiment section), we report GELD’s performance under various search strategies, **including greedy search**. Additionally, we need to clarify that baseline models do not exclusively use greedy search strategies (see the tablenote of Table 1). For example:
> > > - LEHD, BQ, GD: Greedy search.
> > > - Omni-TSP, ELG, UDC: Greedy multiple rollouts.
> > > - INViT-3V: Greedy multiple rollouts with data augmentation.
> > >
> > > Even when restricted to greedy search, without leveraging our innovation in the re-construction process, GELD consistently outperforms all baseline models in terms of inference speed. Furthermore, GELD achieves superior solution quality compared to five models and remains comparable to the other two (UDC and INViT-3V). These results underscore GELD’s robustness and efficiency, even under simpler configurations.
> > >
> > > ---
> > >
> > > ### References
> > >
> > > [1] Graph Neural Network Guided Local Search for the Traveling Salesperson Problem, ICLR 2022.
> > >
> > > [2] Unsupervised Learning for Solving the Travelling Salesman Problem, NeurIPS 2023.
> > >
> > > [3] Position: Rethinking Post-Hoc Search-Based Neural Approaches for Solving Large-Scale Traveling Salesman Problems, ICML 2024.
> > >
> > > [4] Learning to Solve Travelling Salesman Problem with Hardness-adaptive Curriculum, AAAI 2022.
> > >
> > > [5] H-TSP: Hierarchically Solving the Large-Scale Travelling Salesman Problem, AAAI 2023.
> > >
> > > [6] Pointerformer: Deep Reinforced Multi-Pointer Transformer for the Traveling Salesman Problem, AAAI 2023.
> > >
> > > [7] Generalize a Small Pre-trained Model to Arbitrarily Large TSP Instances, AAAI 2021.
> > >
> > > [8] Hierarchical Neural Constructive Solver for Real-world TSP Scenarios, SIGKDD 2024.
> > >
> > > [9] Select and Optimize: Learning to Solve Large-scale TSP Instances, AISTATS 2023.
> > >
> > > [10] A GREAT Architecture for Edge-Based Graph Problems Like TSP, submitted to ICLR 2025.
> > >
> > > [11] MSLC: Monte Carlo Tree Search Sampling Guided Local Construction for Solving Large-Scale Traveling Salesman Problem, submitted to ICLR 2025.
> > >
> > > [12] DDRL: A Diffusion-Driven Reinforcement Learning Approach for Enhanced TSP Solutions, submitted to ICLR 2025.
> > >
> > > [13] Neural Combinatorial Optimization with Heavy Decoder: Toward Large Scale Generalization, NeurIPS’23.
> > >
> > > [14] Self-improvement for neural combinatorial optimization: Sample without replacement, but improvement, TMLR'24.
> > >
> > > [15] DIFUSCO: Graph-based diffusion solvers for combinatorial optimization, NeurIPS’23.
> > >
> > > [16] BQ-NCO: Bisimulation quotienting for efficient neural combinatorial optimization, NeurIPS’23.
> > >
> > > [17] Towards generalizable neural solvers for vehicle routing problems via ensemble with transferrable local policy, IJCAI’24.
> > >
> > > [18] UDC: A unified neural divide-and-conquer framework for large-scale combinatorial optimization problems, NeurIPS’24.

---

> ### Comment · Reviewer_Wg4P · 2024-11-25
> **Thank you for the authors' responses.**
>
> Thank you for the authors' responses.
>
> I am sorry I meant D&C, not D&R in W1. Thank you for pointing it out.
>
> I have read your responses carefully, but I still have some questions:
>
> 1. The authors have mentioned that the biggest contribution/novelty is addressing the lack of a unified method for both producing and improving TSP solutions. Is it necessary to unify the two? Do you have any insights on this?
>
> 2. There are many related works for solving TSP based on D&C [1][2] or heatmap[3], and their work can be applied to various kinds of COPs, including TSP. I still have doubts about the application scope of this paper, especially for ICLR 2025.
>
> 3. The authors list the SOTA algorithms for large-scale TSPs in the experimental part. Actually, I would like to see more analysis, specifically:
> * It would be clearer to list the Heatmap-based and the D&C methods separately, and you can refer to [1].
> * There could be more discussions on the performance of GELD and other Heatmap-based / D&C methods to further claim the advantages of your framework.
> * LKH results should be included in the experiments for comparison.
>
>
> References:
>
> [1] Zheng, Zhi, et al. UDC: A unified neural divide-and-conquer framework for large-scale combinatorial optimization problems. (NeurIPS 2024)
>
> [2] Fu Luo, Xi Lin, Fei Liu, Qingfu Zhang, and Zhenkun Wang. Neural combinatorial optimization with heavy decoder: Toward large scale generalization. (NeurIPS 2023)
>
> [3] DIFUSCO: Graph-based Diffusion Solvers for Combinatorial Optimization. (NeurIPS 2023)

---

> > ### Author Response · Authors · 2024-12-02
> > **Response to Reviewer Wg4P (Round 2, Part 1)**
> >
> > Thank you for your 2nd-round review comments. Here are our detailed responses to your questions.
> >
> > ### Question 1:
> >
> > >The authors have mentioned that the biggest contribution/novelty is addressing the lack of a unified method for both producing and improving TSP solutions. Is it necessary to unify the two? Do you have any insights on this?
> >
> > ### Response 1.5:
> > Thank you for your insightful question. More rigorously, our most significant contribution in this work should be the development of the unified model that functions both as a standalone TSP solver and as an improvement method to **solve TSPs of all sizes**.
> >
> > We believe that unifying neural construction methods (solution generation) and neural improvement methods (solution optimization) is both practically significant and necessary for the following key reasons:
> >
> >    - **Enhanced Flexibility for Real-World Applications**:
> >    A unified model that supports both solution generation and optimization enables efficient adaptation to diverse scenarios, significantly broadening its application scope. For instance, in food delivery route planning, such a model can rapidly generate high-quality routes during peak hours and optimize the routes during off-peak hours for higher solution quality with slightly increased computational time. This flexibility leads to enhanced delivery efficiency and reduced overall costs.
> >
> >    - **Cost and Resource Efficiency**:
> >    Maintaining a single model with a unified set of parameters for both types of tasks reduces resource requirement, training/optimization time, and deployment complexity compared to separate models. This efficiency is particularly valuable in industries with dynamic demands, such as autonomous mining, smart logistic hubs, automated ports and terminals, etc., enabling faster adaptation and streamlined operations.

---

> > > ### Author Response · Authors · 2024-12-02
> > > **Response to Reviewer Wg4P (Round 2, Part 2)**
> > >
> > > ### Comment 2:
> > > >There are many related works for solving TSP based on D&C [1-2] or heatmap[3], and their work can be applied to various kinds of COPs, including TSP. I still have doubts about the application scope of this paper, especially for ICLR 2025.
> > >
> > > ### Response 1.6:
> > > We would like to clarify again that our study focuses exclusively on TSPs to ensure research depth, which coincides with the scope of recent studies published in top-tier conferences such as ICLR, ICML, and NeurIPS (see Response 1.1, Round 1). Specifically, we explore deep learning approaches to solve TSPs of all sizes and propose the following research question based on the four challenges outlined in the Introduction section of our manuscript:
> > > > Can a unified pre-trained model, not based on D&C, effectively solve both small- and large-scale TSPs in a short time period while further elevating solution quality at the cost of affordable time?
> > >
> > > Developing such a unified model has significant practical implications (see Response 1.5, Round 2) but presents substantial challenges. For instance, due to factors such as GPU memory limitations and inference speed, the capability of most neural TSP solvers (including those you mentioned [1-3]) is constrained when solving large-scale TSPs (see Appendix B.5 for details). Moreover, D&C approaches [1] may fail to account for interdependencies between subproblems, which can lead to suboptimal solutions, as we discussed in Appendix B.5 of the revised manuscript. Additionally, we note that the method in [2] is not based on D&C. Moreover, heatmap-based TSP solvers, such as DIFUSCO [3], rely heavily on search strategies such as 2-opt and MCTS, which are often specifically tailored to TSP [4] and can be inefficient for large-scale problems.
> > >
> > > **In contrast, our proposed GELD successfully answers the proposed research question by adopting a novel low-complexity architecture, a two-stage training strategy, and techniques to improve the efficiency of re-construction, supported by extensive experimental results**.
> > >
> > > We acknowledge that using a unified model to effectively solve COPs (not only TSPs) of all sizes while efficiently enhancing solution quality is an exciting and worthwhile direction for future work. However, we believe that achieving this goal requires further research, because these problems involve greater complexity and distinct requirements that **are difficult to address alongside TSP in a single study comprehensively** (like the aforementioned ICLR, ICML, and NeurIPS prior studies of the same problem domain).
> > >
> > > Regarding the application scope of our paper, we are optimistic about the (future) application of GELD to other COPs because prior studies (including those you mentioned [1-3]) have shown that models performing well on TSP can achieve excellent results on other COPs. Notably, our proposed GELD does not rely on operations (such as heatmap + 2-opt or MCTS) that are often specifically tailored to TSP [1]. Moreover, as you recognized in the strength of our work, our proposed Region-Average Linear Attention could be applied in other applications.
> > >
> > > **Given the depth of our research, we believe its scope and quality are well-suited to the prestigious ICLR conference.**

---

> > > > ### Author Response · Authors · 2024-12-02
> > > > **Response to Reviewer Wg4P (Round 2, Part 3)**
> > > >
> > > > ### Comment 3:
> > > > >The authors list the SOTA algorithms for large-scale TSPs in the experimental part. Actually, I would like to see more analysis, specifically:
> > > > It would be clearer to list the Heatmap-based and the D&C methods separately, and you can refer to [1].
> > > > There could be more discussions on the performance of GELD and other Heatmap-based / D&C methods to further claim the advantages of your framework.
> > > > LKH results should be included in the experiments for comparison.
> > > >
> > > > ### Response 1.7:
> > > >
> > > > Thank you for your valuable suggestions. To provide a comprehensive comparative analysis, we selected seven SOTA baseline models, each representing recent advancements (from 2023 to 2024) in the field. Among these, only one is based on the D&C method [1]. The heatmap-based models [3] are discussed separately in Appendix B.6 (see Response 1.4, Round 1). Therefore, we deem that categorizing a single D&C-based model [1] would not be necessary. Instead, we chose to present the baseline models in chronological order across all tables to better highlight the rapid progression of neural TSP solvers, by which we emphasize the timeliness and relevance of our work in relation to the current research frontier.
> > > >
> > > > Per your suggestion to further discuss the advantages of our proposed GELD framework compared to heatmap-based and D&C methods, we have expanded the discussion on the comparisons with a SOTA D&C method UDC and a heatmap-based model DIFUSCO in the revised manuscript (see Appendices B.5 and B.6). Specifically:
> > > >
> > > > 1. >The results on real-world datasets and synthetic datasets demonstrate that GELD outperforms all baseline models, including the SOTA D\&C-based model UDC. This superior performance can be attributed to GELD's effective integration of global and local information, whereas UDC is suboptimal in these experiments because it may overlook correlations between sub-problems.
> > > >
> > > > 2. >We present the comparison results on the synthetic TSP instances of the uniform distribution in Table 10. Notably, the performance of DIFUSCO is heavily dependent on the iterative optimization process of 2-opt that (often) specifically tailored to TSP, while our method does not. As shown, our proposed GELD outperforms DIFUSCO and the nearest neighbor+2-opt heuristic across all problem sizes in terms of both solution quality and computational efficiency.
> > > >
> > > > Regarding the inclusion of LKH results:
> > > >
> > > > As we mentioned in Appendix B.4, the performance of all models, including the baseline and our proposed models, is evaluated relative to LKH (see (15) in Appendix B.4). This allows for an implicit comparison in all the presented tables. Nonetheless, to address your concern more explicitly, we have now presented the LKH results on the synthetic dataset in Table 6 of the revised manuscript. The results are as follows:
> > > > | Distribution | TSP-100 | TSP-500 | TSP-1000 | TSP-5000 | TSP-10000 |
> > > > |--------------|-------------------|-------------------|--------------------|--------------------|---------------------|
> > > > | **uniform**   | 7.8693            | 16.5601           | 23.2215            | 50.9830             | 73.1436             |
> > > > | **clustered** | 5.3876            | 10.3447           | 14.0982            | 28.8359            | 40.2628             |
> > > > | **explosion** | 6.5397            | 12.0101           | 16.0543            | 31.9792            | 41.2801             |
> > > > | **implosion** | 7.1135            | 14.4128           | 20.1932            | 45.0435            | 63.7273             |
> > > >
> > > > We believe that this inclusion provides a clearer comparison and enhances the overall clarity of our experiments.
> > > >
> > > > ---
> > > >
> > > > ### References
> > > >
> > > > [1] UDC: A unified neural divide-and-conquer framework for large-scale combinatorial optimization problems, NeurIPS'24.
> > > >
> > > > [2] Neural Combinatorial Optimization with Heavy Decoder: Toward Large Scale Generalization, NeurIPS'23.
> > > >
> > > > [3] DIFUSCO: Graph-based diffusion solvers for combinatorial optimization, NeurIPS'23.
> > > >
> > > > [4] Self-improved learning for scalable neural combinatorial optimization, arXiv'24

---

> > > > > ### Comment · Reviewer_Wg4P · 2024-12-03
> > > > >
> > > > > I would like to express my gratitude for the hard work the authors have put into their rebuttal.
> > > > >
> > > > > I agree with the assertion that the capacity to address TSPs of varying sizes is occasionally essential in practical applications, and this point deserves significant emphasis within your manuscript. The additional experiments have demonstrated encouraging results of large-scale TSPs.
> > > > >
> > > > > I believe that the paper has the potential for further enhancement by incorporating the feedback from all reviewers. At this stage, I have chosen to increase my rating to 5.

---

> > > > > > ### Author Response · Authors · 2024-12-03
> > > > > > **Official Comment by Authors**
> > > > > >
> > > > > > Thank you for raising your review score. We are glad that we have adequately addressed your concerns.

---

### Meta-Review · Area_Chair_4cZv · 2024-12-17

**Metareview:**

This paper presents GELD, a neural TSP solver aiming to handle TSPs of various scales. It combines a global encoder and local decoder with a novel attention mechanism and a two-stage training strategy. Claims include outperforming state-of-the-art models in solution quality and speed. Strengths lie in its novel architecture elements like the Region-Average Linear Attention mechanism and attempts to address TSPs across a wide range of sizes. However, weaknesses are notable. Experimental aspects have issues such as weak quantity for only focusing on TSP, potential novelty concerns with similar existing concepts, and questions about experimental setup and control. There are also doubts about the claim of handling "All Sizes" as the time complexity's scalability for extremely large instances isn't fully proven. The paper is rejected as despite authors' efforts in rebuttal to address reviewer concerns regarding comparisons, novelty, and experimental details, fundamental issues around the robustness of its claimed broad applicability and some aspects of experimental design remain unresolved, failing to meet the standards for acceptance.

In the reviewer discussion, points were raised about aspects like experimental quantity, novelty of proposed methods, and fairness of experimental setups. The authors responded with explanations on focusing on TSP for depth, clarifications on methodological differences for novelty, and justifications for dataset and comparison choices. But in the final assessment, these responses didn't fully alleviate concerns regarding the core claims and overall experimental soundness, leading to the rejection decision.

**Additional Comments On Reviewer Discussion:**

During the rebuttal period, reviewers brought up multiple concerns. One was the limited scope of experiments only on TSP, which the authors countered by stating it was for research depth and citing precedents of similar focused studies. Regarding novelty questions on methods like PRC, authors detailed how their model differed in functionality and parameterization from related works. For experimental setup issues such as dataset choice and comparison with specific models, authors provided explanations on dataset representativeness and difficulties in including certain models. However, in weighing these points for the final decision, it was determined that while the authors made attempts to address them, key issues around the overall strength of evidence for the main claims and the adequacy of the experimental framework persisted, thus justifying the rejection.

---

### Decision · Program_Chairs · 2025-01-22

Reject